# The transcriptional co-repressor Runx1t1 is essential for MYCN-driven neuroblastoma tumorigenesis

Jayne E. Murray[1,2,14], Emanuele Valli [1,14], Giorgio Milazzo [3,14], Chelsea Mayoh [1,2], Andrew J. Gifford [1,2,4], Jamie I. Fletcher [1,2], Chengyuan Xue[1], Nisitha Jayatilleke [1], Firoozeh Salehzadeh[1], Laura D. Gamble [1], Jourdin R. C. Rouaen[1], Daniel R. Carter [1,2,5], Helen Forgham [1], Eric O. Sekyere[1], Joanna Keating[1], Georgina Eden[1], Sophie Allan[1], Stephanie Alfred[1], Frances K. Kusuma [1], Ashleigh Clark[1], Hannah Webber[1], Amanda J. Russell [1,13], Antoine de Weck [1], Benjamin T. Kile[6,7], Martina Santulli [3], Piergiuseppe De Rosa [3], Emmy D. G. Fleuren[1,2], Weiman Gao[1], Lorna Wilkinson-White[8], Jason K. K. Low [9], Joel P. Mackay [9], Glenn M. Marshall [1,2,10], Douglas J. Hilton[7], Federico M. Giorgi [3], Jan Koster [11], Giovanni Perini [3,15], Michelle Haber [1,2,15] & Murray D. Norris [1,12,15] ✉

*MYCN* oncogene amplification is frequently observed in aggressive childhood neuroblastoma. Using an unbiased large-scale mutagenesis screen in neuroblastoma-prone transgenic mice, we identify a single germline point mutation in the transcriptional corepressor Runx1t1, which abolishes MYCN-driven tumorigenesis. This loss-of-function mutation disrupts a highly conserved zinc finger domain within Runx1t1. Deletion of one *Runx1t1* allele in an independent *Runx1t1* knockout mouse model is also sufficient to prevent MYCN-driven neuroblastoma development, and reverse ganglia hyperplasia, a known pre-requisite for tumorigenesis. Silencing *RUNX1T1* in human neuroblastoma cells decreases colony formation in vitro, and inhibits tumor growth in vivo. Moreover, *RUNX1T1* knockdown inhibits the viability of PAX3-FOXO1 fusion-driven rhabdomyosarcoma and MYC-driven small cell lung cancer cells. Despite the role of Runx1t1 in MYCN-driven tumorigenesis neither gene directly regulates the other. We show RUNX1T1 forms part of a transcriptional LSD1-CoREST3-HDAC repressive complex recruited by HAND2 to enhancer regions to regulate chromatin accessibility and cell-fate pathway genes.

Neuroblastoma, a disease of the neural crest and the most common tumor of infancy, contributes ~15% of pediatric cancer-related deaths, despite accounting for only 7–10% of all childhood cancer cases in developed countries[1,2]. The disease putatively arises from primitive sympathetic neural precursor cells, and tumors can develop at any point along the sympathetic nervous system, although are commonly found to arise within the adrenal medulla[2]. Amplification of the *MYCN* oncogene, critically important for patient stratification, occurs in ~25% of neuroblastomas and can lead to amplification levels greater than 100-fold[2,3]. *MYCN*, a transcription factor and member of the *MYC*

family of proto-oncogenes, is essential for normal neuronal development, with mice lacking *Mycn* dying during embryogenesis with severe defects in the brain, heart, lungs, and central and peripheral nervous systems[4]. *MYCN*-amplified tumors are associated with aggressive disease, and amplification status remains a major prognostic marker used to stratify children to a high-risk category[5]. The survival rate of high-risk patients remains poor despite intensive therapy, and those patients who do survive have long-term sequelae[6]. Alternative treatments with fewer long-term side effects, therefore, are urgently needed.

The *Th-MYCN* neuroblastoma mouse model, involving targeted expression of human *MYCN* to neural crest cells, recapitulates aggressive human neuroblastoma. Tumors are histologically either undifferentiated or poorly differentiated neuroblastoma with a gene expression profile resembling late-stage *MYCN*-amplified disease[7]. Mice homozygous for the *MYCN* transgene all develop neuroblastoma by 7 weeks of age[8,9], making this an ideal model for an unbiased genetic suppressor screen using *N*-ethyl-*N*-nitrosourea (ENU) mutagenesis[10]. ENU induces random, primarily point, mutations into the spermatogonial stem cells of male mice[11], and has proven a powerful tool for elucidating genes and pathways involved in development and disease.

Here, we identify using ENU mutagenesis of *Th-MYCN* mice a pedigree harboring a single loss-of-function mutation in the co-repressor *Runx1t1*, which segregates with abolition of MYCN-driven neuroblastoma tumorigenesis. We also demonstrate this loss-of-function effect in an independent *Runx1t1* knockout mouse model, and via in vitro and in vivo studies silencing *RUNX1T1* in human neuroblastoma cells. These findings were extended to show that loss of RUNX1T1 also inhibits the viability of PAX3-FOXO1 fusion alveolar rhabdomyosarcoma (aRMS) and MYC-driven small cell lung cancer (SCLC) cells. In this work, we show that RUNX1T1 forms part of a repressive complex in neuroblastoma cells acting on enhancer regions to support MYCN and HAND2 in driving neuroblast hyperplasia and eventual transformation.

## Results

### Runx1t1 impacts MYCN-driven neuroblastoma development

To identify genes associated with MYCN-driven tumorigenesis, male *Th-MYCN* homozygous mice (129/SvJ background) were injected with ENU prior to mating with female homozygotes. Of 1716 viable offspring screened, a founder mouse (#1590) developed a tumor at 49 weeks of age (Supplementary Fig. 1a). Histopathological analysis of this tumor showed it to be a teratoma rather than neuroblastoma (Supplementary Fig. 1b). For this model, the lack of neuroblastoma tumor development was unprecedented, where normal tumor development in homozygous mice is invariably observed by 7 weeks of age (Supplementary Fig. 1a). The offspring of #1590 demonstrated equivalent neuroblastoma suppression with distribution showing a dominant Mendelian trait: half developing tumors normally by 7 weeks of age (unsuppressed; *n* = 11) and half displaying a delayed tumor phenotype (suppressed; *n* = 11), with only one of the latter mice developing a tumor (Fig. 1a, b).

To map the chromosomal location associated with this phenotype, the #1590 line was backcrossed to either BALB/c or C57BL/6 mice and bred to congenicity. The suppressed tumor phenotype was retained on both backgrounds when compared to normal *Th-MYCN* mice backcrossed using the same strategy (Supplementary Fig. 1c, d). Whole genome sequencing narrowed the critical region to the proximal end of chromosome 4 (9.4–43.07 megabase region), and further sequencing identified a single thymine:cytosine change in one allele of the *Runx1 partner transcriptional co-repressor 1* (*Runx1t1*) gene at coordinate chr4:13816819 (mm10). This mutation resulted in a tyrosine to histidine amino acid substitution (Y534H) and occurred within nervy homology region 4 (NHR4) of *Runx1t1*, one of four highly conserved regions displaying homology to the *nervy* gene found in Drosophila

(Fig. 1c). Y534 was found to be invariant across species from Drosophila to humans.

NHR4 contains a MYND (**MY**eloid, **N**ervy, and **D**EAF-1) zinc-finger domain, which mediates protein interactions[12]. Protein modeling showed that Y534 is in the core of the MYND domain and suggested that substitution of this tyrosine with histidine would likely alter the nearby zinc coordination centers that are integral to the structure and be highly disruptive to the function of this domain (Fig. 1c).

### Runx1t1 haploinsufficiency prevents MYCN-driven tumorigenesis

To determine if the *Runx1t1*[Y534H] mutation was causing the suppressed tumor phenotype, an independent mouse model with insertional inactivation of *Runx1t1*[13] was crossed with *Th-MYCN* animals. Homozygous knockout of *Runx1t1* (*Runx1t1*[-/-]) resulted in death 1–2 days after birth as previously reported[13]. Loss of one functional allele of *Runx1t1* (*Runx1t1*[+/-]) almost completely abolished tumor development in *Th-MYCN*[+/+] homozygous mice (Fig. 1d). Mice with wild-type *Runx1t1* (*Runx1t1*[+/+]) had 92% tumor incidence (93/101), as expected on this background, while *Runx1t1* haploinsufficiency decreased tumor incidence to 6.5% (10/163). This effect was replicated in mice hemizygous for the *MYCN* transgene, despite these mice having decreased tumor penetrance and longer latency due to a reduced *MYCN* dosage. The *Th-MYCN*[+/-]; *Runx1t1*[+/+] mice displayed 9.9% tumor incidence (18/181), as compared to 0.87% (3/343) in *Th-MYCN*[+/-]; *Runx1t1*[+/-] mice (Supplementary Fig. 1e). This effect was also observed in *MYCN* hemizygous *Runx1t1*[+/Y534H] mice (Supplementary Fig. 1f), suggesting the *Runx1t1*[Y534H] mutation is a loss-of-function in the context of neuroblastoma development, and providing strong evidence for *Runx1t1* having an important role in neuroblastoma tumorigenesis in this model.

### Runx1t1 blocks the maturation of precursor cells of neuroblastoma

We have previously shown that tumor initiation in *Th-MYCN* mice is dependent upon the inappropriate persistence of neuroblast hyperplasia within the paravertebral sympathetic ganglia[9]. Neuroblast hyperplasia is part of normal development, and in this study we observed a variable though generally high proportion of mice at birth exhibit neuroblast hyperplasia within the sympathetic ganglia (mean 52–76%) regardless of genotype (Fig. 2a). In mice lacking the *MYCN* transgene (*Th-MYCN*[-/-]; *Runx1t1*[+/+] or *Th-MYCN*[-/-]; *Runx1t1*[+/-]), these hyperplastic regions regress by 2 weeks of age and mice show no evidence of neuroblastoma formation. However, *Th-MYCN*[+/+]; *Runx1t1*[+/+] mice demonstrated persistence of neuroblast hyperplasia out to four weeks, while those with only one functional *Runx1t1* allele (*Th-MYCN*[+/+]; *Runx1t1*[+/-]) underwent regression of hyperplasia rapidly after birth, and by 4 weeks were indistinguishable from mice lacking the *MYCN* transgene (Fig. 2a). Irrespective of genotype, a small number of Runx1t1 positive neuroblasts were generally observed in the sympathetic ganglia at birth (Fig. 2b; left panels). However, by four weeks of age, *Th-MYCN*[+/+]; *Runx1t1*[+/+] mice showed a marked increase in the number of Runx1t1 positive neuroblasts, while for all other genotypes Runx1t1 expression was virtually absent (Fig. 2b; right panels).

A similar phenomenon was also observed in the *Th-MYCN*[+/+]; *Runx1t1*[+/Y534H] mice, which showed complete regression of neuroblast hyperplasia by four weeks after birth (Supplementary Fig. 2a). In brain samples obtained from *Th-MYCN*[+/+] mice with either wild-type or mutant *Runx1t1*, Runx1t1 protein expression was not significantly different between both groups, indicating that the point mutation does not affect Runx1t1 protein stability (Supplementary Fig. 2b). These results suggest a critical window between 0-4 weeks where Runx1t1 is required for neuroblastoma formation.

MYCN is believed to exert its effects via a block to normal cell differentiation pathways such as neuritogenesis. Consistent with this,

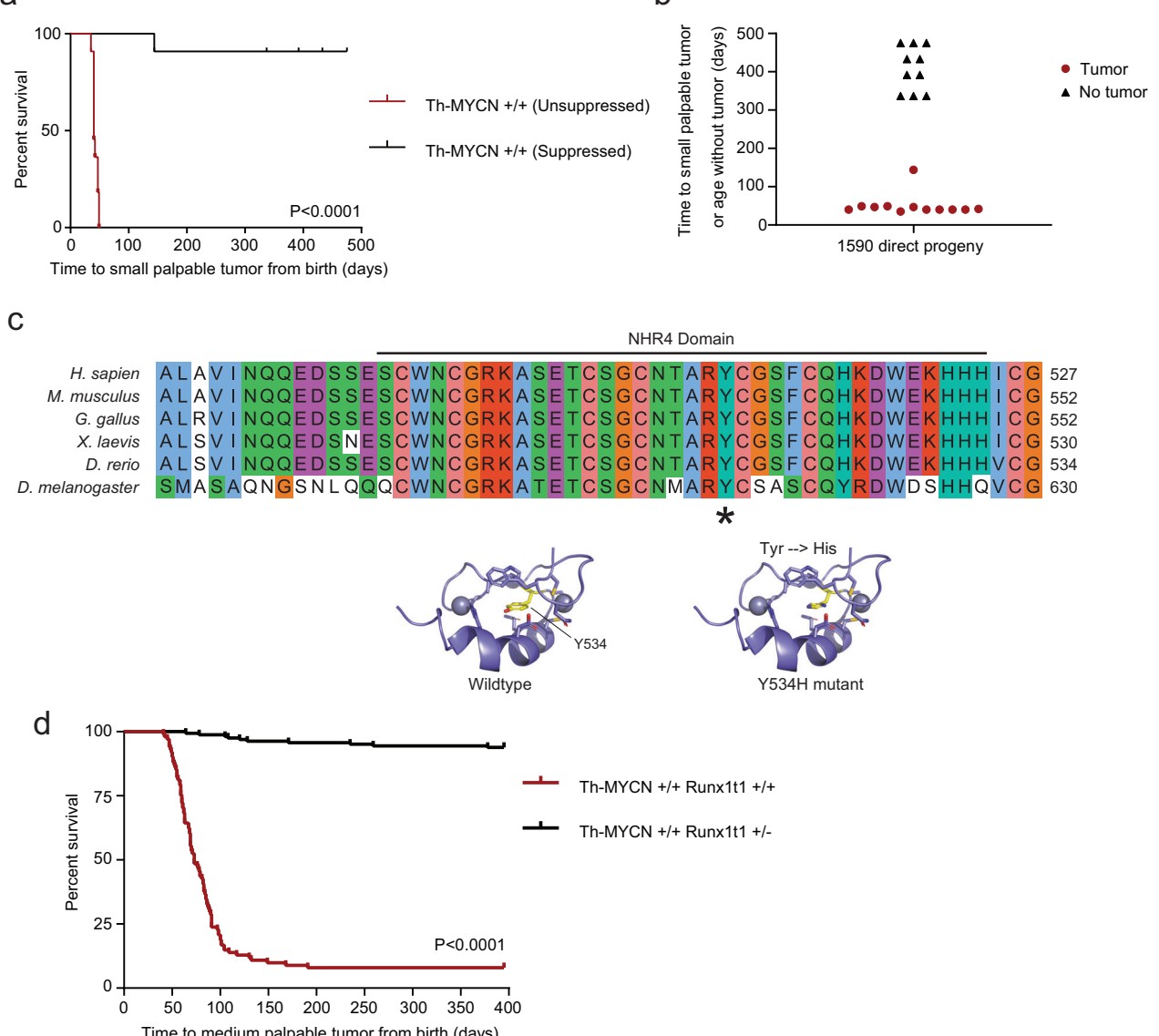

**Fig. 1 | Single point mutation in NHR4 domain of *Runx1t1* that abrogates neuroblastoma development. a** Kaplan–Meier survival analysis of progeny of the ENU-treated *Th-MYCN* mouse #1590, that were either wild-type for *Runx1t1* (red, unsuppressed) or had inherited the Y534H *Runx1t1* mutation (black, suppressed); $n = 11$ per genotype, $p < 0.0001$ (log-rank (Mantel–Cox) test). **b** Scatter plot showing time to tumor development in progeny of the ENU-treated mouse #1590 that were wild-type for *Runx1t1* (red), or had inherited the Y534H mutation (black), $n = 11$ per genotype. **c** Multiple protein sequence alignment of the RUNX1T1 NHR4 domain across a range of organisms. The suppressed tumor phenotype resulted from the substitution of histidine (H) for a highly conserved tyrosine residue (Y), denoted by the asterisk (upper panel). A schematic model of wild-type and mutant Runx1t1 NHR4 zinc-finger motif domain folding (bottom panel). **d** Kaplan–Meier survival analysis for homozygous *Th-MYCN* mice either wild type (*Th-MYCN +/+ Runx1t1 +/+*) or with heterozygous knock-out (*Th-MYCN +/+ Runx1t1 +/−*) of *Runx1t1*. Wild-type *Runx1t1* mice (red) demonstrated almost complete tumor penetrance, while *Runx1t1* heterozygous knock-out mice (black) almost entirely lacked the ability to form tumors ($n = 101$ for wild-type and $n = 163$ for heterozygous knock-out, log-rank (Mantel–Cox) test $p < 0.0001$). Source data are provided as a Source Data file.

in vitro culture of whole ganglia from *Th-MYCN⁺/⁺; Runx1t1⁺/⁺* mice demonstrated a lack of neurite outgrowth by comparison with either *Th-MYCN⁻/⁻* mice or *Th-MYCN⁺/⁺; Runx1t1⁺/⁻* mice (Fig. 2c). Furthermore, RNAseq analysis showed that ganglia from transgenic mice with wild-type *Runx1t1* (*Th-MYCN⁺/⁺; Runx1t1⁺/⁺*) clustered more closely with fully developed tumors from these mice, while ganglia from *Th-MYCN* mice with only one functional copy of *Runx1t1*, clustered with *Th-MYCN* mice lacking the *MYCN transgene*, regardless of their *Runx1t1* genotype (Fig. 2d). These findings indicate that neuroblastoma tumorigenesis in this transgenic model requires a sustained high level of Runx1t1 above that needed for normal embryonal development.

## MYCN drives increased RUNX1T1 translation

Although high levels of *Runx1t1* RNA were found in the brains and tumors of *Th-MYCN* mice with wild-type *Runx1t1* compared to liver tissue, high levels of corresponding *MYCN* RNA were observed only in the tumors of these mice (Fig. 3a). In contrast, high protein levels of both MYCN and Runx1t1 were only observed in tumor samples (Fig. 3b). Furthermore, examination of human neuroblastoma cell lines showed no significant difference in the mean level of *RUNX1T1* RNA expression between *MYCN*-amplified and non-amplified lines (Fig. 3c), while RUNX1T1 protein levels were markedly higher in those lines with amplified *MYCN* (Fig. 3d), suggesting that MYCN drives increased RUNX1T1 translation, not transcription.

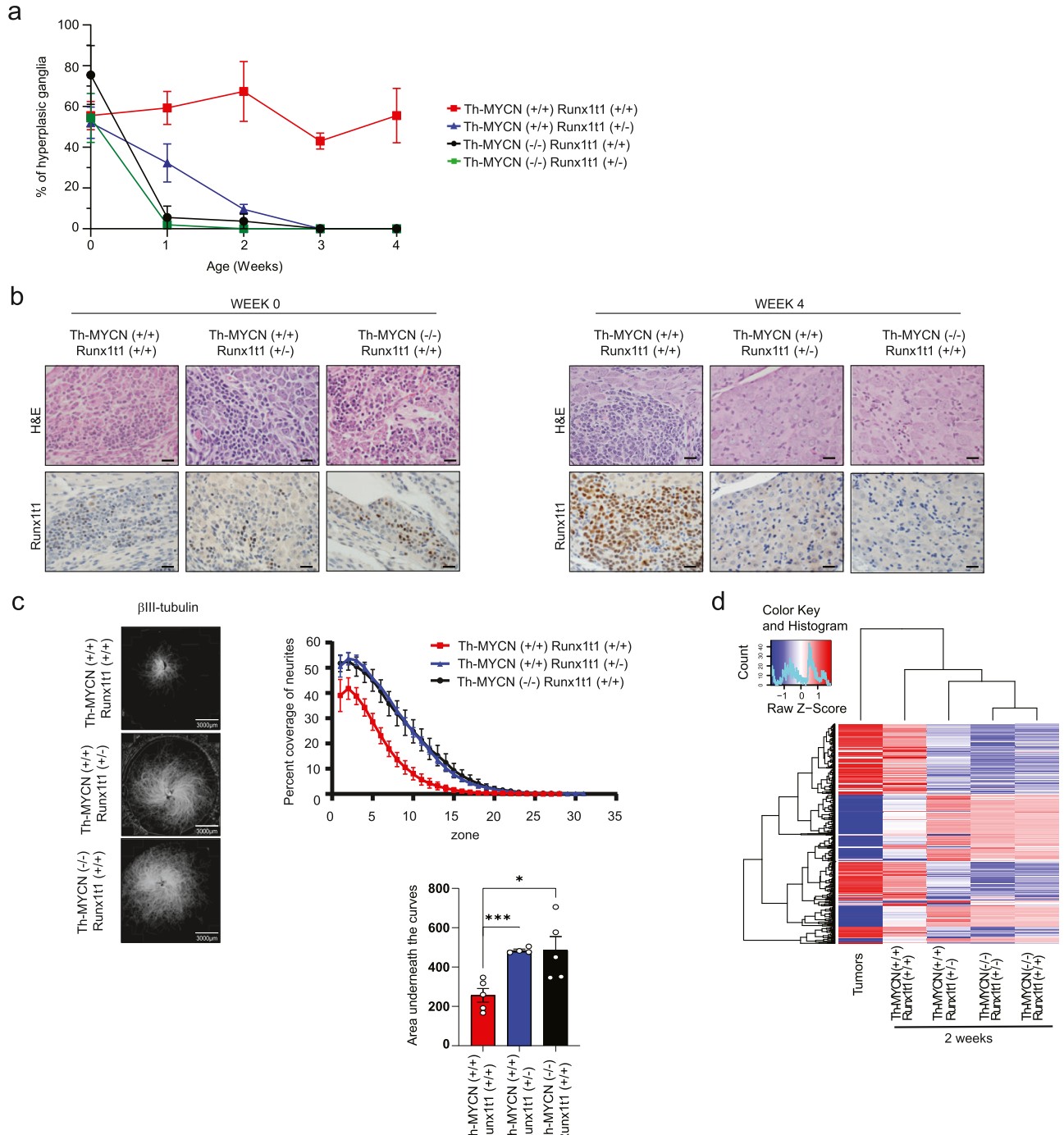

**Fig. 2 | Runx1t1 loss reverses MYCN-mediated sustained hyperplasia and induces ganglia neurite extension. a** The percentage neuroblast hyperplasia scored from homozygous *Th-MYCN* (+/+) mice or littermate mice lacking the *MYCN* transgene (−/−), with either wild-type (+/+) or heterozygous loss (+/−) of *Runx1t1*. Scoring of N = 3–8 independent mice was performed for each genotype and timepoint. All data points were N = 3, except for +/+, +/+ week 1 and week 2 (N = 4); +/+, +/− day 0 (N = 8) and week 4 (N = 4); −/−, +/− day 0 (N = 6), week 1 (N = 4) and week 4 (N = 5). The graph is mean ± SEM. **b** Representative histology of RUNX1T1 staining in ganglia from mice homozygous for the *Th-MYCN* transgene, and either wild-type or heterozygous for *Runx1t1* from day 0 and 4 weeks of age.

Neuroblast hyperplasia is defined as ≥30 small round blue cells within a sympathetic ganglion[9]. Photos were taken at 600X magnification, and the scale bars represent 20 microns. **c** βIII-tubulin staining of sympathetic ganglia isolated from *Th-MYCN* mice. The percent coverage of neurites was calculated and the area under the curve determined for each ganglia. N = 4 (+/+, +/−) or 5 (−/−, +/+ and +/+, +/+), Graphs are Mean ± SEM, *p = 0.0181, ***p = 0.0007. Two-tailed unpaired *t*-test. **d** Heatmap displaying gene clustering following RNA-Seq analysis of ganglia dissected from two-week-old *Th-MYCN* mice with either homozygous or heterozygous *Runx1t1*, compared to fully developed murine neuroblastoma tumors. Source data are provided as a Source Data file.

To investigate the association of *RUNX1T1* with patient outcome in primary neuroblastoma, Kaplan–Meier survival analysis was performed on a publicly available RNAseq dataset of 498 neuroblastoma samples. Unexpectedly, low expression rather than high expression

of *RUNX1T1* mRNA was found to be predictive of poor outcome (Supplementary Fig. 2c). In addition, the level of *RUNX1T1* mRNA was significantly lower in *MYCN*-amplified tumors by comparison with non-amplified tumors (Fig. 3e). Given these results, we next analysed

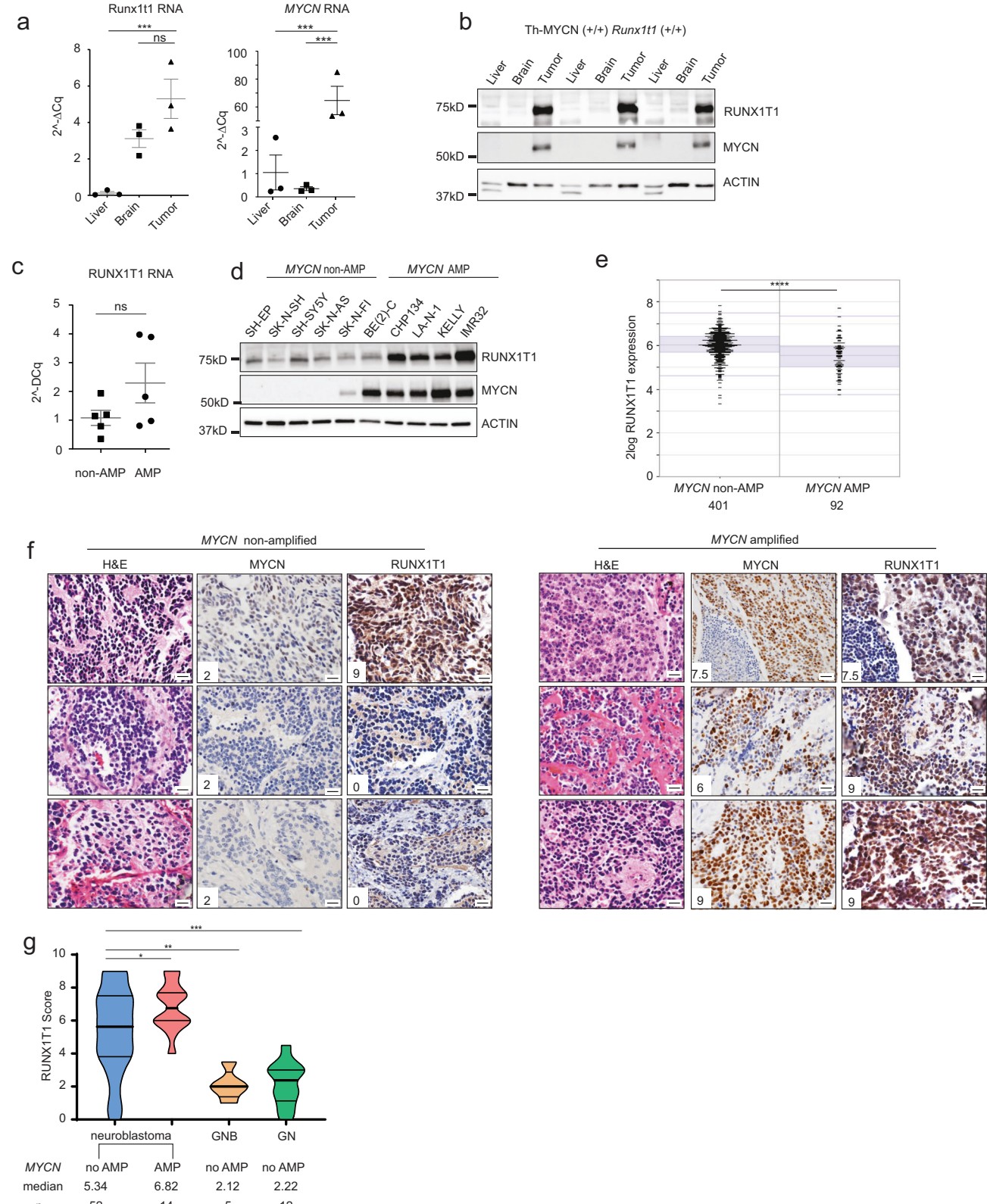

RUNX1T1 protein expression in an independent tissue microarray (TMA) of primary neuroblastoma ($n = 77$), ganglioneuroblastoma ($n = 5$), and ganglioneuroma ($n = 12$). The neuroblastoma TMA cohort has been shown to be representative of neuroblastoma in general[14], and widespread strong RUNX1T1 nuclear staining ("high" expression) was observed in both *MYCN*-amplified and non-amplified neuroblastoma samples (Fig. 3f; Representative images shown). Following

quantification, *MYCN*-amplified tumors had significantly higher RUNX1T1 protein levels than non-amplified samples, with low levels of RUNX1T1 protein only observed in non-*MYCN*-amplified tumors. Neuroblastoma samples as a group had higher RUNX1T1 levels compared to the more benign forms of ganglioneuroblastoma and ganglioneuroma (Fig. 3g). In contrast to the results observed with *RUNX1T1* mRNA, higher RUNX1T1 protein also correlated with poorer

**Fig. 3 | RUNX1T1 protein level correlates with neuroblastoma aggressiveness.**
**a** *Runx1t1* and *MYCN* mRNA expression in liver, brain, and tumor samples from homozygous *Th-MYCN* mice with intact *Runx1t1* relative to the control, *Gusb*. Means ± SEM (*n* = 3) are indicated. Data were analysed using one-way-ANOVA and corrected for the multiple comparison using the Tukey test. ***p* = 0.0042, ns *p* = 0.1385 for *Runx1t1*; ****p* = 0.0006 for *MYCN*. Three independent mice were used for each tissue. **b** Western blot for RUNX1T1 and MYCN in liver, brain, and tumor from homozygous *Th-MYCN* mice; *n* = 3 independent mice. **c** *RUNX1T1* mRNA expression relative to the control *GUSB* in *MYCN*-non-amplified (non-AMP) and amplified (AMP) neuroblastoma cell lines. Means ± SEM (*n* = 5 data points representing five different cell lines in each group, and the value of each data point is the mean of three biological repeats) are indicated, two-tailed Mann–Whitney test, *P* = 0.3095. **d** Western blot for RUNX1T1 and MYCN across a panel of *MYCN* non-amplified (non-AMP) and *MYCN* amplified (AMP) neuroblastoma cell lines. This experiment has been repeated once with similar results. **e** Scatter plot of *RUNX1T1* mRNA expression in *MYCN* non-amplified (non-AMP) and amplified (AMP) tumors from a publicly available dataset (SEQC) of tumor samples from the neuroblastoma R2 database, *n* = 410 non-AMP and *n* = 92 AMP samples; two-tailed unpaired *t*-test, *****p* < 0.0001. **f** A tissue microarray (TMA) of human neuroblastoma tumor samples (*n* = 66) was stained with antibodies to either MYCN or RUNX1T1. Photos from three representative *MYCN*-non-amplified samples and three *MYCN*-amplified samples are shown. Numbers in panels indicate staining intensity. Photos were taken at 600× magnification, and the scale bars represent 20 microns. H&E, hematoxylin and eosin. **g** RUNX1T1 staining intensity was scored for all samples in the TMA. Neuroblastoma samples were split into non-amplified (*n* = 52) and amplified (*n* = 14) and compared to benign diseases of ganglioneuroblastoma (GNB) (*n* = 5) and ganglioneuroma (GN) (*n* = 12). Violin plot describes distribution of intensity, width describes frequency of score in tumors. The thick line represents the median, thin line represents the 25th and 75th percentiles. The bounds of the box show the range of scores. Unpaired *t*-test, two-tailed **p* = 0.0403; ***p* = 0.0064; ****p* = 0.0001. Source data are provided as a Source Data file.

clinical outcome (Supplementary Fig. 2d). These results provide evidence for MYCN driving increased RUNX1T1 protein translation in neuroblastoma cells.

To determine if RUNX1T1 is regulated by MYCN, we analysed RUNX1T1 protein and RNA expression in the human SH-EP Tet-21/N cell line, a non-amplified neuroblastoma line with a *MYCN* doxycycline-off inducible construct. RUNX1T1 protein levels were significantly decreased after doxycycline treatment, whereas no significant change was detected in the mRNA level, despite achieving significant decreases of both MYCN protein and RNA (Fig. 4a, b). In addition, forced overexpression of MYCN in SH-EP cells (SH-EP MYCN) resulted in upregulation of RUNX1T1 protein (Fig. 4c), with no change in the levels of mRNA (Fig. 4d), suggesting either increased RUNX1T1 translation or protein stabilization in the presence of high levels of MYCN. Cycloheximide chase assays performed on two *MYCN*-amplified (BE(2)-C and KELLY) and one non-amplified (SH-EP) neuroblastoma lines, revealed no significant differences in protein stability (Fig. 4e). However, transfection of a luciferase vector containing the 5′UTR of the most abundant murine *Runx1t1* isoform into SH-EP cells with or without MYCN overexpression, revealed significantly increased luciferase activity from this 5′UTR in the presence of MYCN overexpression, indicating an increased level of translation (Fig. 4f). Together, these results show that rather than direct transcriptional regulation of *RUNX1T1*, MYCN is driving increased protein translation.

## Loss of RUNX1T1 downregulates MAX protein levels
The finding of high levels of RUNX1T1 in *MYCN*-amplified cells due to increased protein translation is in keeping with the well-characterized role of MYCN in driving increased ribosome biogenesis[15]. However, shRNA-mediated silencing of RUNX1T1 did not lead to a corresponding decrease in MYCN protein but rather an increase (Supplementary Fig. 3a), indicating that RUNX1T1 affects the activity of MYCN and not its transcription or protein stability. Unlike other transcription factors, MYC proteins are rarely mutated in cancers, and oncogenic transformation most often results from aberrant expression of a normal protein[16]. MYCN cannot exert its function without dimerizing with its obligate partner MAX, and this partnership is necessary for driving cell growth and inhibiting differentiation[17]. In contrast to MYCN, which has a short half-life in normal cells and undergoes rapid degradation, MAX is highly stable in both resting and proliferating cells[18]. We have previously shown that high-level MAX expression in *MYCN*-amplified primary neuroblastomas confers a poor outcome, while knockdown of MAX in *MYCN*-amplified cell lines inhibits both cell growth and motility, suggesting that MAX has a critical role in MYCN-mediated oncogenesis[19]. Here, we observed a >50% reduction in the level of MAX protein following RUNX1T1 depletion in KELLY cells (Supplementary Fig. 3b). MAX also dimerizes with members of the MXD family, which are known to antagonize the activation of genes by MYC/MAX.

However, we observed either undetectable protein or no significant change in levels of MXD1-7 following RUNX1T1 knockdown (Supplementary Fig. 3c).

We next performed co-immunoprecipitation (Co-IP) with MYCN and MAX and showed that the lower MAX protein level observed following RUNX1T1 depletion also led to a reduction in the ratio MAX:MYCN binding by comparison with control cells (Supplementary Fig. 3d). This, in turn, led to significant downregulation of the MYCN target genes *GSPT1*, *CLNS1A* and *NAP1L1* (Supplementary Fig. 3e). These results provide evidence that RUNX1T1 depletion leads to inhibition of MYCN target gene transcription via MAX downregulation.

## RUNX1T1 loss inhibits neuroblastoma proliferation
To explore whether RUNX1T1 plays a role in the maintenance of established neuroblastomas, we generated cell lines with doxycycline-inducible shRNA knockdown of this co-repressor. *MYCN*-amplified KELLY (Fig. 5a, c) and BE(2)-C (Fig. 5b, d), as well as non-amplified SH-SY5Y cells (Supplementary Fig. 4a, b), all displayed reduced clonogenic capacity when RUNX1T1 was downregulated after doxycycline treatment. Xenografting KELLY and BE(2)-C cell lines into immunocompromised mice followed by doxycycline treatment resulted in significantly increased survival for both lines (Fig. 5e, f), compared to control mice. Immunostaining of KELLY tumor samples excised at seven days confirmed loss of RUNX1T1 following its knockdown, as well as decreased cell proliferation as indicated by a reduced Ki67 proliferative index (47.5 ± 4.4% in knockdown tumors compared to 80.4 ± 4.3% in control tumors; *P* = 0.029) (Fig. 5g, left panels). A small reduction in Ki67 staining remained in KELLY xenografts at the endpoint following RUNX1T1 knockdown (65.0 ± 2.1%) compared to control tumors (71.8 ± 1.6; Fig. 5g, right panels). In contrast, there was no reduction in the level of MYCN at either of these timepoints (Supplementary Fig. 4c). No overall marked morphological differences were observed in KELLY tumors with RUNX1T1 knock-down and control tumors at either Day 7 or endpoint (Supplementary Table 1). In both cohorts, the tumors were either undifferentiated or poorly differentiated neuroblastoma and most tumor cells were viable. BE(2)-C cells showed reduced levels of RUNX1T1 in doxycycline-treated samples obtained at the endpoint of the experiment compared to untreated samples. The degree of reduced expression, however, was greater for RUNX1T1 than MYCN (Supplementary Fig. 4d). Together, these results highlight a role for RUNX1T1 in tumor initiation and in vivo neuroblastoma cell growth.

## RUNX1T1 forms part of an LSD1, CoREST, HDAC repressor complex
RUNX1T1 acts as a transcriptional co-repressor by interacting with DNA binding basic helix–loop–helix (bHLH) transcription factors (TCF3, TCF4, TCF12), co-repressor associated proteins (SIN3A, NCOR1, NCOR2/SMRT) and histone deacetylases (HDAC1, HDAC2, HDAC3). In

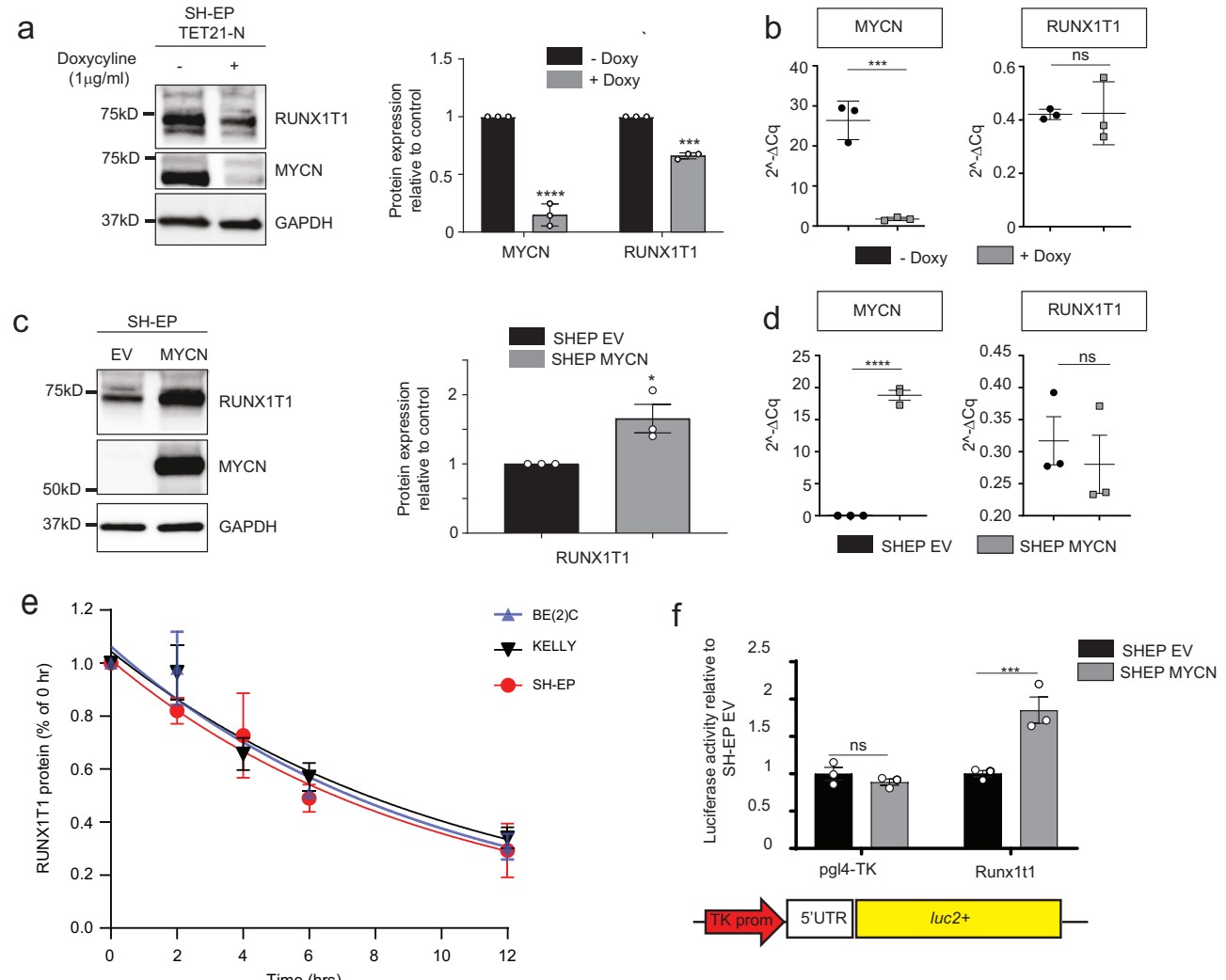

**Fig. 4 | MYCN drives increased RUNX1T1 translation. a** Western blot of SH-EP TET-21/N cells after 72 h of treatment with 1 μg/mL doxycycline (left panel). Quantitation from three independent experiments demonstrated significantly decreased MYCN (****$p = 0.0001$) and RUNX1T1 expression (***$p < 0.0001$) (right panel). Values represent means from three independent experiments ±SD, two-tailed unpaired *t*-test. **b** *MYCN* and *RUNX1T1* mRNA expression after 72 h doxycycline treatment, showing significant downregulation of *MYCN* (***$p = 0.0009$) but no significant change in RUNX1T1 ($p = 0.9559$). Values represent means from three independent experiments ±SD, two-tailed unpaired *t*-test. **c** Western blot of SH-EP neuroblastoma cells overexpressing MYCN and empty vector (EV) control (left panel). Quantitation of western blot, from three independent experiments (right panel) demonstrated significantly increased RUNX1T1 expression (*$p = 0.0339$).

Values represent means from three independent experiments ±SD, two-tailed unpaired *t*-test. **d** *RUNX1T1* and *MYCN* mRNA expression in SH-EP cells overexpressing MYCN, relative to the control gene (*GUSB*). Values represent means from three independent experiments ±SD. Two-tailed unpaired *t*-test, ****$p < 0.0001$; ns: not significant ($p = 0.5679$). **e** Cyclohexamide chase **e**xperiment showing RUNX1T1 protein stability in three neuroblastoma cell lines (BE(2)-C, KELLY, SH-EP) over a 12 h time course. Values represent means from three independent experiments ±SD. **f** Luciferase activity following transfection of murine *Runx1t1* 5'UTR into MYCN overexpressing or EV SH-EP cells. Values represent means from three independent experiments ±SD. Groups were analysed using two-way ANOVA and Bonferroni multiple comparison. ***$p = 0.0007$; ns not significant ($p = 0.9335$). Source data are provided as a Source Data file.

particular, the NHR4 zinc finger domain is responsible for specifically mediating interactions with NCOR co-repressors and HDACs[20–22]. To assess the structural impact of the Y534H mutation on RUNX1T1 function, wild-type and mutant RUNX1T1 MYND domains were cloned into bacterial expression vectors and their conformation examined by one-dimensional [1]H NMR spectroscopy. In contrast to the wild-type domain, which demonstrated signals that were sharp and well dispersed (indicative of a well-folded domain), the mutant domain gave rise to a poor-quality spectrum that exhibited considerable peak broadening, suggesting either nonspecific aggregation or oscillation between many partially ordered states, likely to cause a loss-of-function (Fig. 6a).

To define the RUNX1T1 protein interactome in neuroblastoma cells, we performed mass spectrometry (LC-MS/MS) on BE(2)-C cells transiently transfected with FLAG-tagged wild-type RUNX1T1 followed

by STRING analysis to define putative RUNX1T1 interacting partners (Supplementary Fig. 5a). RUNX1T1 has been reported to interact with components of a CoREST chromatin-remodeling complex, containing both HDAC repressive activity and de-methylating LSD1 enzymatic activity[23,24], as well as bHLH transcription factors and we observed similar findings (Supplementary Fig. 5a). Many of these proteins have been shown to have a role in neuroblastoma or neuronal development[25–28].

Co-IP validated the interaction between wild-type RUNX1T1 and the transcription factor HAND2 as well as the key CoREST factors LSD1/KDM1A, RCOR3/CoREST3, and HDAC1-3 (Fig. 6b, c). However, with the introduction of the Y534H (YH) point mutation, this interaction was reduced for HDAC3, and was completely lost with CoREST3 (Fig. 6c). Individual deletion of each of the four RUNX1T1 NHR regions

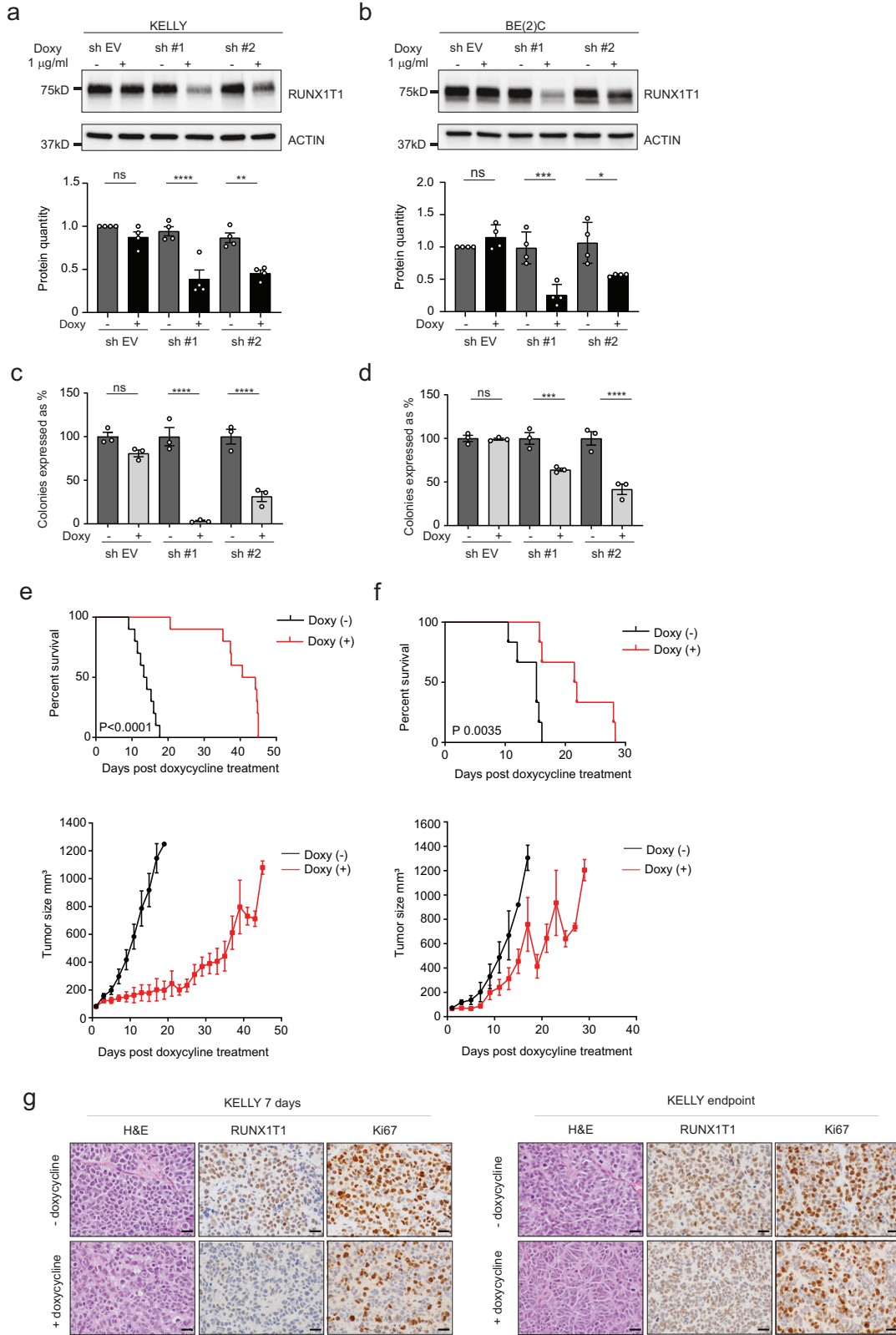

confirmed the dependence of the RUNX1T1-CoREST3 interaction on the presence of the NHR4 domain (Supplementary Fig. 5b, c). Conversely, individual deletion of each of the four RUNX1T1 NHR regions showed the independence of the NHR4 domain for the RUNX1T1-LSD1 interaction (Supplementary Fig. 5d). Furthermore, we identified three motifs within the C-terminal region of CoREST3 with high homology to a similar motif (PPPLIS) within NCOR2/SMRT (Fig. 6d) that is known to

bind to the NHR4 MYND domain[12]. Using NMR titrations, peptides containing two of these CoREST3 motifs were also found to bind to the NHR4 MYND domain although with a two-to-three-fold weaker affinity than NCOR2. These experiments were carried out at pH 6 to allow direct comparison with NMR data for NCOR2 binding to the NHR4 MYND domain from ref. 12. The affinity of the best binding peptide was also assessed at pH 7.5 under buffer conditions that allowed

**Fig. 5 | *RUNX1T1* knockdown reduces neuroblastoma cell proliferation both in vivo and in vitro. a, b** RUNX1T1 knock down after 72hrs of doxycycline (1 μg/mL) treatment in KELLY and BE(2)-C cells respectively, with two independent shRNA constructs. Quantitation from four independent experiments demonstrated significantly decreased RUNX1T1 levels following shRNA-mediated knockdown. Values represent means from three independent experiments ±SD. Ordinary one-way ANOVA with Tukey's multiple comparisons test; KELLY sh#1 ****$p < 0.0001$, sh #2 **$p = 0.0018$, ns $p = 0.6962$; BE(2)-C sh #1 ***$p = 0.0006$, sh #2 *$p = 0.0295$, ns $p = 0.8753$. **c, d** Colony formation after doxycycline-induced knockdown of RUNX1T1 in KELLY and BE(2)-C cells respectively, with two independent shRNA constructs. Colonies are represented as a percentage relative to the untreated control in each experiment. Values represent means from three independent experiments ±SD. RUNX1T1 shRNA-mediated knockdown resulted in significantly decreased colony numbers compared to controls. Ordinary 1-way ANOVA with

Tukey's multiple comparisons test; KELLY sh #1 ****$p < 0.0001$, sh #2 ****$p < 0.0001$, ns $p = 0.3589$; BE(2)-C sh #1 ***$p = 0.0005$, sh #2 ****$p < 0.0001$, ns $p = 0.8549$. **e, f** Kaplan–Meier survival analysis of NSG mice xenografted with doxycycline-inducible *RUNX1T1* shRNA#1 in KELLY and BE(2)-C cells, respectively. Mice were split into two groups ($n = 10$ mice per group for KELLY and $n = 6$ mice per group for BE(2)-C) and fed doxycycline or control food once a 50 mm³ tumor was measurable. $P$ values were determined using the Log-Rank (Mantel–Cox) test, $p < 0.0001$ for KELLY and $p = 0.0035$ for BE(2)-C. Growth curves (lower panels) plot size over time, post-doxycycline treatment. Graphs are Mean ± SEM. **g** Representative images of KELLY cells at 7 days and endpoint ± doxycycline. Tumor samples were stained with H&E or immunohistochemically for RUNX1T1 and Ki67. Photos were taken at 600× magnification, and the scale bars represent 20 microns. Source data are provided as a Source Data file.

comparison with ITC data collected by ref. 12 for NCOR2 (Fig. 6d). Based on these data, two CoREST3 mutant proteins, one lacking an amino acid sequence of the last PPPLI motif (Δ1) and another lacking all three "PPPLI motifs" identified (Δ2), were generated (Fig. 6e) and Co-IP performed with RUNX1T1. Although a decrease in binding to RUNX1T1 was observed with the CoREST3_Δ1-mutant, binding of the CoREST3_Δ2-mutant protein to RUNX1T1 was completely lost (Fig. 6e), suggesting that this region of CoREST3 is critical for the binding to RUNX1T1. Overall, these results demonstrate that RUNX1T1 is part of an LSD1, CoREST3, HDAC (LCH) repressor complex and that CoREST3 and RUNX1T1 directly interact, specifically through their PPPLI motif and MYND domain, respectively.

## HAND2 recruits RUNX1T1 to enhancer regions

To map the genomic binding sites of RUNX1T1 and explore the observed interaction between RUNX1T1, HAND2 and the LCH complex, we performed chromatin immunoprecipitation and sequencing (ChIP-seq) on KELLY cells. RUNX1T1 binding occurred almost exclusively within intergenic regions of the genome, rather than gene promoters (Fig. 7a, b). Of the 14019 peaks common to the RUNX1T1 ChIP-seq replicates, a de novo motif search using HOMER revealed several highly significantly enriched motifs, the top ones of which aligned with known motifs for HAND2, PHOX2A, and TCF4 (Fig. 7c). All three of these genes have been shown to have roles in neural crest specification or neuronal development[29–31]. Furthermore, 36% of RUNX1T1 ChIP-seq peaks were found to overlap with HAND2 ChIP-seq peaks (5073) previously reported in KELLY cells[26], suggesting that HAND2 likely recruits RUNX1T1 to specific genomic locations (Fig. 7d). Additionally, we performed ChIP-qPCR analysis for RUNX1T1 following shRNA-mediated inducible knockdown of HAND2. Loss of HAND2 had no significant effect on RUNX1T1 expression (Supplementary Fig. 5e, f). However, ChIP-qPCR performed on four randomly chosen ChIP-seq peaks previously found to be positive for both HAND2 and RUNX1T1, demonstrated a significant decrease in RUNX1T1 binding following the loss of HAND2 (Supplementary Fig. 5g). We, therefore, performed additional ChIP-seq for LSD1/KDM1A and CoREST3, and as anticipated observed a high level of co-localization of RUNX1T1 peaks (41% and 43%, respectively) with these two chromatin modifiers, while 24% of all RUNX1T1 peaks co-localized with HAND2, LSD1, and CoREST3 (Fig. 7d). These findings are in marked contrast with those for the MYCN ChIP-seq dataset from KELLY cells[26], where 2.6% of RUNX1T1 ChIP-seq peaks overlapped with MYCN peaks, while only 1.6% of RUNX1T1 peaks co-localized with MYCN, HAND2, LSD1, and CoREST3 (Fig. 7d). We next used Genomic Regions Enrichment of Annotations Tool (GREAT) analysis to explore gene ontology processes associated with the 24% of common peaks co-localizing with RUNX1T1, HAND2, LSD1, and CoREST3. Highly significant associations were observed with neuronal differentiation and development, including autonomic nervous system development, sympathetic nervous system development, noradrenergic neuron differentiation, nerve development, and neural crest cell migration (Fig. 7e).

We also generated ChIP-seq datasets with histone markers (H3K4me1, H3K4me3, H3K27me3, and H3K27ac) to distinguish poised (H3K4me1 + H3K27me3 marks), primed (H3K4me1 mark only) and active enhancer regions (H3K4me1 + H3K27ac marks). Downregulation of RUNX1T1 led to an increase in the number of regions associated with active enhancers as well as an increase in the number of primed enhancer regions, while there was no observed change in the small number of poised enhancer regions (Fig. 7f). Altogether, these data provide evidence for recruitment by HAND2 of a RUNX1T1 repressor complex to help maintain an undifferentiated phenotype in *MYCN*-amplified neuroblastoma cells.

## RUNX1T1 inhibition reverses an ESC-like signature

Neuroblastoma is widely regarded as a tumor with embryonal origins[32]. There are many similarities between cancer development and embryogenesis with numerous poorly differentiated human cancers exhibiting an embryonic stem cell-like (ESC-like) gene expression signature that includes increased expression of Myc target genes and silencing of Polycomb Repressive Complex 2 (PRC2) target genes[33]. We performed RNAseq analysis in KELLY cells following knockdown of *RUNX1T1* with two individual shRNAs in three separate experiments (shRNA1 in vitro and in vivo; shRNA2 in vitro). Gene set enrichment analysis (GSEA) on the Hallmark gene sets, showed significant downregulation of MYC targets genes (MYC_Targets_V1) as well as of MYC-controlled fundamental processes of cell cycle progression (G2M_checkpoint; E2F_targets) and metabolism (oxidative_phosphorylation; fatty_acid_metabolism) (Fig. 8a). MYC target genes and oxidative phosphorylation Hallmark sets were highly significantly downregulated three to five-fold in all three shRNA experiments (Fig. 8b and Supplementary Fig. 6a). On the other hand, MYC oncoproteins are known to drive suppression of tumor immunity[34,35], and the top upregulated Hallmark gene sets showed enrichment of IL2_STAT5_signaling, TNFA_signaling_via_NFKB and Hypoxia (Fig. 8a), all of which have roles in inflammation and immunity[36–38]. Furthermore, when we included in this analysis RNAseq data of ganglia (*Th-MYCN^{+/+}; Runx1t1^{+/−}*) and tumor (*Th-MYCN^{+/+}; Runx1t1^{+/+}*) from our neuroblastoma-prone mice, GSEA showed highly concordant results with the human data, with enrichment of the same top upregulated and downregulated Hallmark gene sets in mouse ganglia carrying only one function *Runx1t1* allele (Fig. 8a).

Epigenetic changes are considered to be a fundamental hallmark of pediatric tumors including neuroblastoma[39,40], and previous studies have shown that MYCN is able to interact with and recruit PRC2 to epigenetically suppress target genes, a process that appears to be essential for MYCN oncogenesis[41–44]. GSEA showed significant enrichment and increased expression of PRC2 target genes (Benporath_PRC2_Targets) following depletion of RUNX1T1 (Fig. 8c). Target gene enrichment was also observed for the PRC2 subunits EED and SUZ12, as well as genes characteristic of ESCs with the repressive

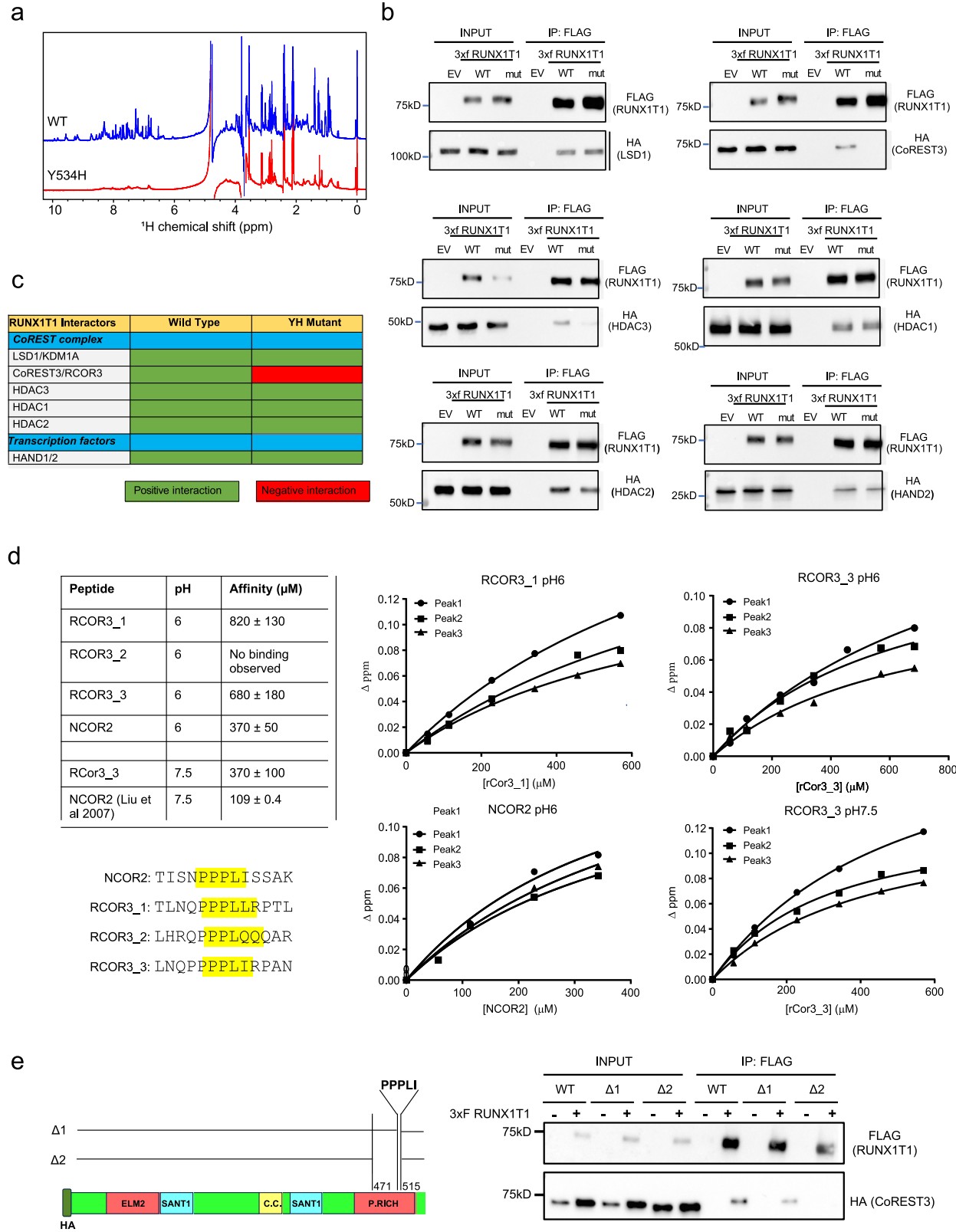

H3K27me3 promoter mark (Supplementary Fig. 6b). In addition, high levels of the MYC target gene signature and low levels of the PRC2 target gene signature were associated with particularly poor outcome in 498 primary neuroblastoma tumors (Supplementary Fig. 6c). Altogether, these results provide evidence for RUNX1T1 being required to maintain a MYCN-driven ESC-like phenotype characteristic of aggressive *MYCN*-amplified neuroblastoma.

## RUNX1T1 depletion impacts alveolar rhabdomyosarcoma and SCLC cells

Interrogation of the Cancer Cell Line Encyclopedia[45] showed that neuroblastoma, SCLC, and sarcoma had the highest levels of *RUNX1T1* expression compared to all other tumor types (Supplementary Fig. 7a). Within the sarcoma subtypes, childhood rhabdomyosarcoma showed the highest levels of *RUNX1T1*. Both rhabdomyosarcoma and SCLC

**Fig. 6 | RUNX1T1 forms part of an LSD1, COREST, and HDAC repressor complex and binds RCOR3 via its NHR4 domain. a** One-dimensional $^1$H NMR spectra of the Runx1t1 MYND domain showed that the wild-type spectrum (top) has sharp peaks and good dispersion whereas the spectrum of the Y534H mutant (bottom) indicates a protein not well-ordered and/or aggregated—consistent with the mutation disrupting proper folding of the domain. **b** Co-IP between RUNX1T1_WT/YH-3xFLAG and LSD1-HA (top left), CoREST3-HA (top right), HDAC3-HA (middle left), HDAC1-HA (middle right), HDAC2-HA (bottom left), and HAND2-HA (bottom right). HEK-293T cells were transiently co-transfected with pCMV14 RUNX1T1_WT- or Y534H-3xFLAG and HA-tagged constructs. Nuclear fractions were immunoprecipitated with anti-FLAG antibody and immunoblotted with both anti-FLAG antibody (IP) and anti-HA (Co-IP). An experiment was performed twice. The input sample represents 5% of the non-immunoprecipitated sample and the control sample (EV) resulted from the co-transfection with the HA-tagged construct and the pCMV14 empty vector. **c** Summary of the Co-IP results. **d** Binding curves and calculated affinities for peptides RCOR3_1-3 and NCOR2 at pH 6 and RCOR3_3 and NCOR2 at

pH 7.5. Binding curves were derived by tracking the combined $^1$H and $^{15}$N Chemical Shift Perturbations of three signals in the MYND domain $^{15}$N-HSQC following titration of each peptide in increments of 0.5 molar equivalents into samples of $^{15}$N-HNR4 MYND, each of which was fitted to a simple 1:1 binding model. The affinities reported for each peptide are the average (±SD) of three sets of measurements. We estimate the uncertainty for the KD's to be ~25%. The affinity of NCOR2 at pH 7.5 was calculated by ref. [12] and is listed for comparison. **e** Schema of CoREST3/RCOR3 deletion mutants (left). HEK-293T cells were transiently co-transfected with pCMV14 RUNX1T1_WT-3xFLAG and HA-tagged CoREST3 constructs (right, $n = 1$ experiment). Nuclear fractions were immunoprecipitated with anti-FLAG antibody and samples immunoblotted and probed with anti-FLAG antibody (IP, top boxes), or anti-HA (Co-IP, bottom boxes). Input represents 5% of non-immunoprecipitated sample; −/+ indicates co-transfection of pCMV14 EV and HA-CoREST3 constructs (−) or co-transfection of pCMV14 RUNX1T1_WT-3xFLAG and HA-CoREST3 constructs (+). Source data are provided as a Source Data file.

frequently display amplification of one of the MYC family gene members, including *MYCN*[46,47]. Two alveolar rhabdomyosarcoma (aRMS) lines, Rh41 and Rh3, which are t(2;13) PAX3-FOXO1 fusion-positive and express high levels of RUNX1T1, demonstrated significantly reduced cell proliferation following shRNA knockdown (Fig. 8d). We also performed knockdown with two independent *RUNX1T1* siRNAs in the Rh41 cells and over a 96 hour period and again demonstrated significant growth inhibition (Supplementary Fig. 7b).

A recent report has also shown amplification as well as high expression of *RUNX1T1* in SCLC cells by comparison with non-SCLC[48]. We observed high-level RUNX1T1 expression in five SCLC cell lines by comparison with the non-small cell lung cancer A549 cells and SH-SY5Y neuroblastoma cells (Supplementary Fig. 7c). Furthermore, shRNA-mediated *RUNX1T1* downregulation led to a significant decrease in colony number in SCLC cells in MYC-driven DMS-273 and DMS-53 (Fig. 8e and Supplementary Fig. 7d, e). These results suggest that RUNX1T1 plays an important role in the malignant phenotype of MYC/MYCN-driven aRMS and SCLC and is a potential therapeutic target.

## Discussion

The finding that RUNX1T1 depletion upregulates PRC2 target genes and downregulates MYC targets corresponds with the available evidence indicating that MYCN physically interacts with PRC2 and recruits this complex to epigenetically repress gene expression[44,49,50]. Furthermore, the critical role of PRC2 in embryonal development is demonstrated by mouse knockouts of the core components of this complex (Ezh2, Eed, and Suz12), which are all embryonic lethal[51], while a CRISPR-Cas9 screen performed in *MYCN*-amplified neuroblastoma cells demonstrated a top dependency on EZH2, EED, and SUZ12[43]. Expression of EZH2, a transcriptional target of MYCN, is associated with neuroblastoma outcome, and inhibition of this histone methyltransferase by genetic or pharmacologic means, leads to neuroblastoma growth inhibition[43,44,52–54]. A recent report has also shown that many neuroendocrine cancers, including neuroblastoma, use PRC2 to repress MHC-I antigen-processing pathway (MHC-I APP) genes, thereby evading immune surveillance[55]. Thus, inhibition of RUNX1T1 by reversing the effects of PRC2 may similarly restore anti-tumor immunity in this disease and further studies of this using immunocompetent models are warranted.

Our data showing that RUNX1T1 depletion leads to the downregulation of MAX underscores the role that MAX plays in MYCN-driven tumorigenesis. MYC:MAX dimerization is a known critical factor for MYC oncogenesis, as demonstrated in Eμ-*Myc*-driven lymphomagenesis, where MAX loss results in complete abrogation of tumor development[56]. A recent study has also shown that although MAX inactivation can accelerate SCLC progression, this effect is independent of MYC, and MAX deletion in MYC-overexpressing SCLC also leads to tumor abrogation[57]. As a result of such findings, there has been

a major focus on developing inhibitors targeting the MYC:MAX interaction, with one promising cell-penetrating peptide (OMO-103) currently undergoing clinical trial (NCT04808362). A recently identified potent MYC:MAX interaction inhibitor (MYCMI-7), was found to downregulate MYC and E2F pathways and upregulate inflammatory signaling via NFκB[58], and these were also the top downregulated and upregulated Hallmark pathways respectively, that we identified following RUNX1T1 depletion.

Previous studies have identified core regulatory circuitries in high-risk neuroblastoma cells involving a discrete set of master transcription factor genes associated with super-enhancers that are responsible for the maintenance of cell state[25,26]. Although we could find no evidence from ChIP-seq data indicating the binding of RUNX1T1 to super-enhancer regions, we found a high level of overlap of RUNX1T1 ChIP-seq peaks with previously reported HAND2 ChIP-seq peaks[26]. HAND2, a key component of these core regulatory circuits, has an essential role in neurogenesis and neural crest cell specification, particularly neural crest-derived noradrenergic sympathetic neuron development[30,59]. This master transcription factor forms part of a feed-forward loop that enhances the sensitivity of the neurons to nerve growth factor, and hence their survival, by upregulating the expression of *NTRK1/TrkA*[59]. Our RNAseq data showed *NTRK1* to be one of the most highly upregulated genes following shRNA-mediated silencing of *RUNX1T1*. The RUNX1T1 Y534H mutation led to reduced interaction with HAND2 further suggesting that HAND2 recruits RUNX1T1 to a co-repressor complex to maintain an undifferentiated phenotype in neuroblastoma. We, therefore, propose a model in which *MYCN* amplification drives high-level MYCN activity and an ESC-like phenotype that is supported by increased translation of RUNX1T1 (Fig. 8f). High levels of RUNX1T1 in turn, lead to enhancer-mediated repression of HAND2 target genes required for neuronal differentiation. However, loss of RUNX1T1, through either normal regulation or loss of function mechanisms, indirectly leads to inhibition of MYCN activity via a decreased level of MYCN dimerization partner MAX, and, in addition, upregulation of HAND2-driven pro-differentiation genes.

RUNX1T1 was shown to be part of an LCH repressor complex involving demethylase (LSD1/KDM1A), RCOR3/CoREST3, and histone deacetylase (HDAC1-3) proteins. LSD1 plays a major role in neuronal differentiation and high levels are associated with poorly differentiated neuroblastoma tumors and confer a poor outcome[60]. Depending on cell context, LSD1 can have opposing functions, by recruitment to either co-activator or co-repressor complexes to act on target genes[61]. There are three known CoREST proteins, all of which interact with LSD1[62], however, only CoREST3 was found to bind RUNX1T1, and this interaction was lost following mutation of the NHR4 domain. A previous investigation of megakaryocytic differentiation found that whereas CoREST1 and CoREST2 facilitated LSD1-mediated nucleosomal demethylation, CoREST3 antagonized this process[62]. This

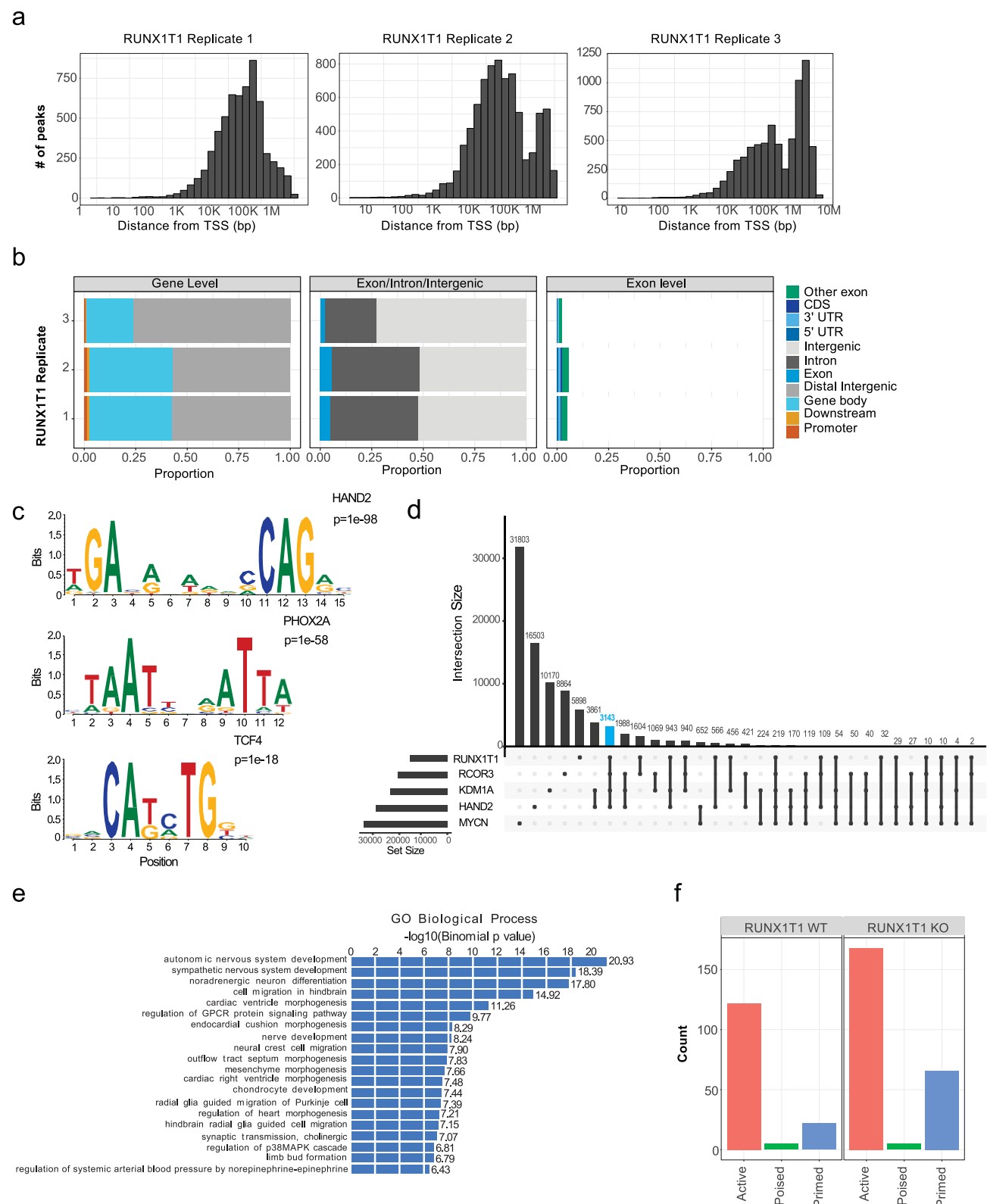

finding suggests that RUNX1T1 downregulation in neural crest cells, leads to loss of interaction with CoREST3 and its antagonistic effects on CoREST1-2, promoting neuronal cell differentiation.

It is known that mouse strain background can influence neuro-blastoma development. Homozygous *Th-MYCN* mice on a 129/SvJ background have 100% tumor penetrance compared to only 5% for the same transgenic mice on a C57BL/6 background, suggesting that 129/SvJ mice possess a strain-specific modifier that enables the *MYCN* transgene

to be more penetrant compared to other strains[63]. Weiss and colleagues mapped a primary tumor susceptibility locus in 129/SvJ mice to chromosome 10, as well as a secondary susceptibility locus to chromosome 4[64]. This latter locus harbors *Runx1t1* and interestingly, the expression of *Runx1t1* was upregulated in 129/SvJ mice compared to another strain of mice (FVB) with a reduced tumor penetrance. Taken together, these findings implicate *Runx1t1* as a potential modifier responsible for the increased tumor penetrance in 129/SvJ *Th-MYCN* mice.

**Fig. 7 | RUNX1T1 binds to intergenic regions and interacts with HAND2 to maintain an undifferentiated neuroblastoma phenotype. a** ChIP-seq analysis of RUNX1T1 binding across three replicates showing the number of peaks and their distance in base-pairs (bp) to the transcription start site (TSS). **b** Proportion of the peaks occurring at different binding site locations from the RUNX1T1 three ChIP-seq replicates. RUNX1T1 binding occurring at either a promoter, gene body, distal intergenic or downstream in relation to a gene (left), binding in an exon, intron or intergenic (middle), and binding within exons at either a coding region (CDS), 5′ untranslated region (UTR) or 3′ UTR (right). **c** Motif discovery using homer analysis

showing significantly enriched motifs from HAND2, PHOX2A, and TCF4. **d** UpSet plot of the intersection of ChIP-seq peaks of RUNX1T1, HAND2, LSD1, and RCOR3. The number of peaks identified are indicated for each gene and the intersection of gene peaks. RUNX1T1 peaks that co-localized with HAND2, LSD1, and CoREST3 are shown by the blue bar. **e** GREAT analysis (using Binomial test) showing enriched gene ontology (GO) biological process of the common peaks between RUNX1T1, HAND2, LSD1, and RCOR3. **f** Number of active, poised, and primed enhancer sites where RUNX1T1 binds in the presence (wildtype; WT) and absence of RUNX1T1 (knock out; KO).

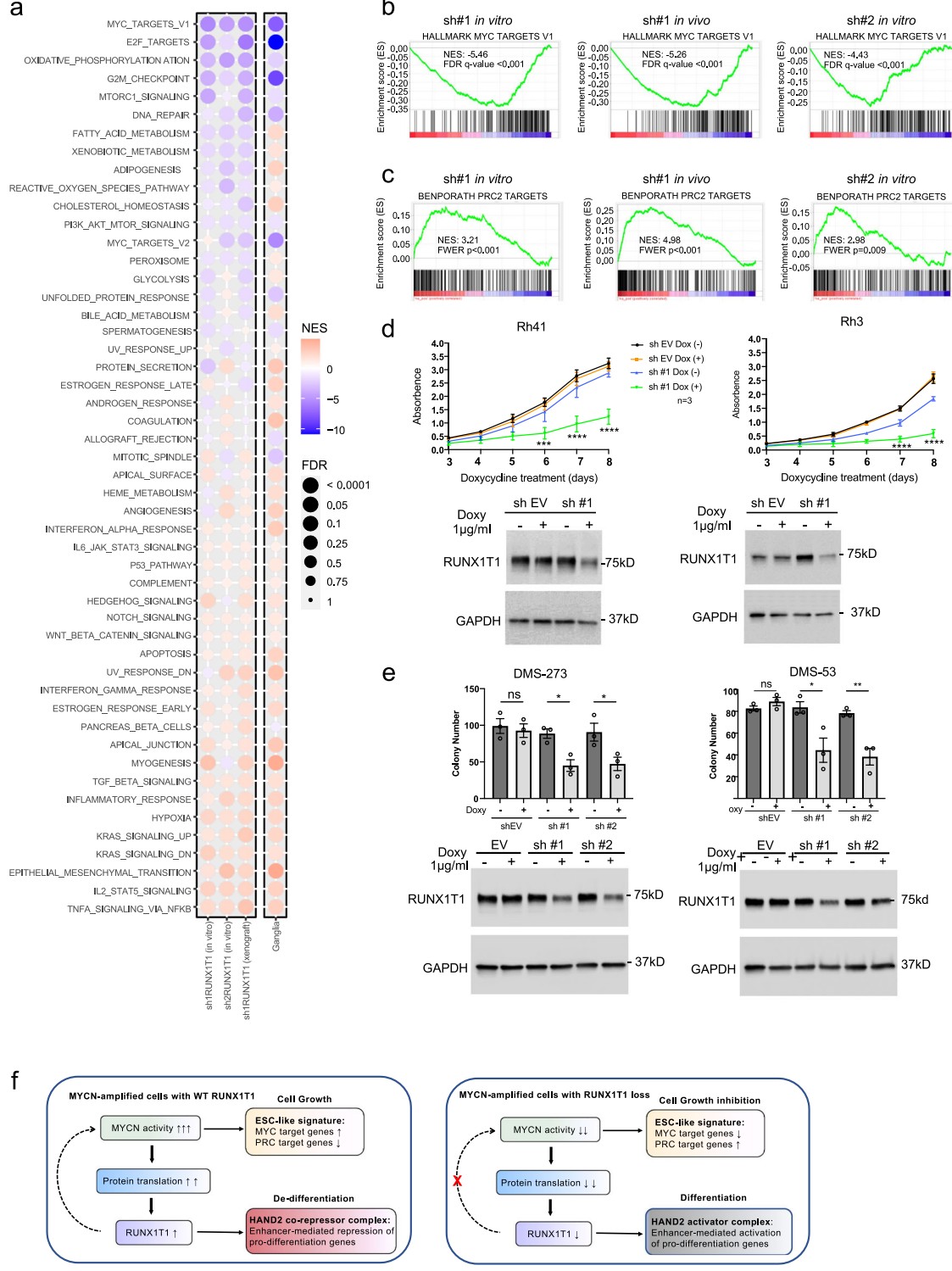

**Fig. 8 | RUNX1T1 depletion downregulates MYCN target genes, upregulates PRC2 silenced genes in neuroblastoma, and inhibits colony formation in aRMS and SCLC cells. a** Gene set enrichment analysis (GSEA) for Hallmark gene sets performed on RNAseq data obtained following *RUNX1T1* shRNA knockdown in KELLY cells. A bubble plot is shown with the size of the bubble representing the significance based on the nominal p value. NES: normalized enrichment score showing the strength and enrichment direction. The column labeled ganglia represents the same analysis performed on ganglia (Th-MYCN$^{+/+}$; Runx1t1$^{+/-}$) at 2 weeks versus tumor obtained from Th-MYCN$^{+/+}$; Runx1t1$^{+/+}$ mice. **b** GSEA plot showing Hallmark MYC_target_V1 genes following *RUNX1T1* shRNA downregulation in KELLY cells. **c** GSEA showed significant enrichment of PRC2 target genes (Ben-Porath_PRC2_targets) following *RUNX1T1* shRNA downregulation in KELLY cells. **d** *RUNX1T1* shRNA downregulation in aRMS cell lines (Rh41 and Rh3) significantly decreased cell proliferation, $n = 3$ independent experiments; graph is mean ± SEM, two-way ANOVA with Dunnett's multiple comparison. Rh41 sh #1 ± Doxy 6d ***$p = 0.0003$, 7d and 8d ****$p < 0.0001$; Rh3 sh #1 ± Doxy 6d *$p = 0.0149$, 7d and 8d ****$p < 0.0001$. The immunoblots show decreased RUNX1T1 levels following shRNA-

mediated knockdown. Each experiment has been repeated three times with similar results. **e** *RUNX1T1* shRNA downregulation in SCLC cell lines (DMS-273 and DMS-53) significantly decreased clonogenic capacity, $n = 3$ independent experiments, graph is mean ± SEM, unpaired two-tailed *t*-test. DMS-273 EV ns $p = 0.6682$, sh#1 *$p = 0.0127$, sh#2 *$p = 0.0465$; DMS-53 EV ns $p = 0.2210$, sh#1 *$p = 0.0325$, sh#2 **$p = 0.0075$. The immunoblots show decreased RUNX1T1 levels following shRNA-mediated knockdown. Each experiment has been repeated three times with similar results. **f** Proposed mechanism of action of RUNX1T1 in *MYCN*-amplified neuro-blastoma. High levels of MYCN resulting from gene amplification drive increased protein translation which includes increased RUNX1T1 protein levels. High-level RUNX1T1 expression is necessary to maintain an ESC-like phenotype as well as generate enhancer-mediated repression of HAND2 targets genes otherwise required for neuronal differentiation. Loss of RUNX1T1 allows upregulation of HAND2-driven pro-differentiation genes while at the same time inhibiting MYCN activity by decreasing the level of the obligate MYCN dimerization partner MAX. Source data are provided as a Source Data file.

Based on our findings, we hypothesized that RUNX1T1 also plays a role in other MYC-driven cancers. PAX3-FOXO1 fusion-positive aRMS tumors frequently display increased expression of MYCN, and it has been shown that this fusion protein orchestrates an epigenetic chromatin-remodeling program in aRMS cells involving the establishment of super enhancers to drive the master transcription factors MYCN and MYOD1[65]. It has also been shown that MYCN can functionally replace MYC during murine development, indicating that the two genes have similar physiological roles and that individual differences likely relate to their transcriptional regulation[66]. He and colleagues reported *RUNX1T1* amplification in the SCLC compartment of patients with mixed tumor histology of both SCLC and non–non-small-cell lung cancer[48]. Our own results showing reduced colony formation in SCLC lines with shRNA-mediated silencing of *RUNX1T1* provide evidence that this co-repressor does function to support this malignant phenotype. In addition, Runx1t1 was identified as one of only 6 transcription factors necessary to reprogram committed murine blood cells to hematopoietic stem cells[67], while a recent study in acute myeloid leukemia (AML) demonstrated that both *Mycn* and *Runx1t1* were able to reprogram multipotent progenitor cells and overcome self-renewal limiting capacity, which in turn led to AML transformation in cells that had lost JARID2, a component of PRC2 and a tumor suppressor[68]. These authors concluded that inhibition of Runx1t1 and MYCN could be exploited as a therapeutic strategy for secondary AML.

In conclusion, our study indicates that Runx1t1 is essential for MYCN-driven tumorigenesis and has provided insight into the key role of this co-repressor not only in neuroblastoma initiation, but also in the progression of this childhood disease, and has opened avenues for the development of potential additional therapeutic treatment approaches.

## Methods

### Animal experiments

The *Th-MYCN* (Tg(Th-MYCN)41Waw, 129/SvJ *Ter* backcross (ARC, Perth, Australia)) mouse model of neuroblastoma overexpresses human *MYCN* in the neuroectodermal cells, and mice develop neuroblastoma as a result[7]. The mice have been maintained by breeding hemizygous mice together. The *Runx1t1* heterozygous knock-out mouse model[13] (CBB6-Runx1t1$^{tm1Fc}$/H) was thawed from sperm at the Medical Research Council Harwell Animal Facility (Oxfordshire, United Kingdom). Animals were imported into Australia (Australian BioResources (ABR), Moss Vale, Australia), passed quarantine, and were shipped to the Children's Cancer Institute Animal Facility. Mice were maintained by breeding heterozygous mice together and were crossed with the *Th-MYCN* mouse. All animals with neuroblastoma were monitored until the specified timepoint or when a 10 mm diameter abdominal tumor was detected, or signs of a thoracic tumor, and this tumor size was not exceeded. Both sexes were used in subsequent experiments.

Female NOD SCID gamma (NOD.Cg-Prkdc$^{scid}$IL2rg$^{tm1Wjl}$/SzJAusb) (ABR, Moss Vale, Australia), aged 5–6 weeks were engrafted sub-cutaneously with cells containing shRNA in the dorsal flank, with 10 million cells for KELLY in 50% Matrigel® matrix (Corning) and 5 million cells for BE(2)-C. Tumors were measured with vernier calipers every second day and the tumor volume was calculated using the formula (length × width × depth)/2. The mice were placed onto food containing 600 mg/kg doxycycline (or control) (Specialty Feeds, Glen Forrest, WA, Australia) when tumors reached between 50–100 mm$^3$ in size and were euthanized when tumors reached 1000 mm$^3$, our maximum approved tumor burden and was only exceeded when overnight growth of tumors went beyond 1000 mm$^3$. For timepoint analysis, tumors were measured for 7 days post-doxycycline treatment and snap-frozen for further analysis.

All mice were housed in a specific pathogen-free, Physical Containment level 2 facility, with the temperature controlled between 22–24 °C and a 12-h day/night. Mice were kept in Ventirack cages (Tecniplast), provided food and water ad libitum, with environmental enrichment. Mice are tail-tipped after birth for genotyping using Chelex® 100 resin as previously described[69]. Real-time PCR sequence-specific assays were designed using the Primer Express™ software, Version 3.0.1 (Applied Biosystems, Thermo Fisher Scientific, Australia), based on the conventional PCR sequences[7,13,70]. Primers and probes for each genotyping assay are listed in Supplementary Table 2. The genotypes of mice were determined using an allelic discrimination methodology with assays designed to the specific region or insertion of disruption or the single point mutation.

All mouse studies were approved by the University of New South Wales Animal Care and Ethics Committee (ACEC) (Approval numbers ACEC 07/58B, 10/8B, 12/97A, 14/122B, 17/14B, 19/89B), according to the Animal Research Act, 1985 (New South Wales, Australia) and the Australian Code of Practice for Care and Use of Animals for Scientific Purposes (2013). The use of genetically modified mice was approved by the University of New South Wales Institutional Biosafety Committee.

### Generation of ENU mutant mice

Homozygote male *Th-MYCN* mice were treated with 30 mg/kg Cyclophosphamide (Baxter) for 5 days upon presentation of a 5 mm tumor at ~5–6 weeks of age. Two weeks after the final injection, the mice were treated with three once weekly injections of 66 mg/kg *N*-ethyl-*N*-nitrosourea (ENU) (Sigma), as previously described[71,72]. Approximately 4 weeks after the last injection of ENU, the male mice were mated with cyclophosphamide-treated female homozygote *Th-MYCN* mice to produce first-generation (G1) progeny. Offspring of both sexes were monitored by abdominal palpation for tumor development. Delayed G1 tumor mice were identified and mated for

heritability to produce G2 offspring. The founder #1590 mouse was identified, and progeny mice were used in all subsequent experiments. For genetic mapping, delayed mice were mated to C57BL/6 (C57BL/6JAusb) or BALB/c (BALB/cJAusb) mice from ABR and then crossed back to non-ENU *Th-MYCN* homozygous mice. Control mice were mated in the same fashion to determine natural tumor drift on a mixed background.

Exome sequencing on #1590 direct progeny was performed at the Australian Phenomics Network located at the Australian National University (Canberra, Australia) using the NimbleGen SeqCap EZ mouse exon array (Roche, California, USA), according to the manufacturer's instructions. The reads were aligned and filtered, removing false positives or negatives where possible and filtering out variants that are not ENU-induced with software. Whole genome sequencing in Chromosome 4 was performed on the backcrossed mice using polymerase chain reaction (PCR) and sequenced on an ABI automatic sequencer according to the manufacturer's instructions (Applied Biosystems, Foster City, CA).

### Immunohistochemistry and tissue microarray

Human neuroblastoma tumor microarray (TMA) sections were obtained from the Tumor Bank at The Children's Hospital at Westmead (The Sydney Children's Hospital Network). Slides were stained with haematoxylin and eosin (H&E), or for RUNX1T1 (rabbit polyclonal antibody, 15494-1-AP, Proteintech, 1:400), MYCN (mouse monoclonal antibody, NCM II 100; ab16898, Abcam, 1:200) by the Garvan Institute of Medical Research Histopathology facility at The Kinghorn Cancer Center (Sydney, Australia). After filtering, 94 tumor cores could be used, comprising 77 neuroblastoma, 5 ganglioneuroblastoma, and 12 ganglioneuroma samples. Clinical outcome data were available for these 94 patients, however 15 patient samples lacked *MYCN* amplification status. Tumors were scored for MYCN and RUNX1T1 by a pediatric pathologist blinded to the clinical outcome. A score was generated for each sample by assessing the percentage of positive staining (0–3, where 0 = 0%, 1 = 1–10%, 2 = 11–50%, 3 = 51–100%) and multiplying this by the intensity of the staining (0–3, 0 = negative, 1 = weak, 1.5 = weak-moderate, 2 = moderate, 2.5 = moderate-strong, 3 = strong). The Ki67 proliferative index (%) was determined by visually estimating the % of viable tumor cells with Ki67 nuclear staining, to the nearest 10%, in tissue microarray cores. If multiple cores were available for the same sample, the values were averaged.

Samples for histology were collected in 10% neutral buffered formalin (Fronine) and fixed for 24 h. Samples were then transferred to 70% ethanol. Skin, teeth, and limbs were removed, and the spines bisected longitudinally. Specimens containing bone, including day 0 pups, were decalcified in 0.38 M EDTA for 48 h. Samples were embedded in paraffin, sectioned, and stained with H&E or for MYCN (rabbit polyclonal antibody, 10159-2-AP, Proteintech, 1:1000) and RUNX1T1 (as described above). Scoring of neuroblast hyperplasia within the autonomic ganglia of mice was undertaken in a blinded study. The percentage of hyperplasia within ganglia scored was then determined. Hyperplasia was defined as ≥30 neuroblasts within a ganglion, as previously described[9,73]. At least three independent mice were scored per genotype per timepoint. Photos were taken using an Olympus BX53 light microscope and DP-73 camera using cellSens software.

### Ganglia RNAseq and whole culture

Celiac ganglia were dissected from 2-week-old *Th-MYCN* homozygote and wild-type mice with 1- or 2-copies of *Runx1t1* and cleaned to remove non-ganglion tissue. Total RNA was isolated using the RNeasy Micro kit (50), #74004 kit, and RNA quality was assessed prior to being used (Agilent 2100 Bioanalyzer, Agilent Technologies). Only RNA with an RNA Integrity Number (RIN) of >9.0 was used. Whole transcriptome sequencing was performed at the Ramaciotti Center for Genomics with a SMARTer Stranded Total RNAseq v2 preparation kit. Prepared

libraries were pooled and sequenced on NovaSeq 6000 S1 2x100bp lane generating an average of 60 million reads per sample. Raw FASTQ files were quality control checked using FASTQC (v0.11.5). Paired-end FASTQ files were aligned to the Mouse genome assembly (build mm10) using STAR (v2.5) two-pass with quantMode parameter set to TranscriptomeSAM. Alignments were sorted and indexed with SAMTools (v.1.3.1). Aligned bam files were processed using RSEM (v1.2.31) to calculate raw gene counts and transcripts per kilobase million (TPM). All RNAseq relative expression values are expressed as TPM values. Differential expression analysis was conducted in R using the edgeR package. Preranked GSEA analysis was performed using MSigDB (v7.4).

For whole culture, superior cervical ganglia were dissected from 2-week-old mice as previously described[9]. Individual ganglia were cultured in Neurobasal-A medium supplemented with B27 (Life Technologies Australia) and antibiotics (penicillin/streptomycin) in 6-well or 12-well plates precoated sequentially with poly-D-lysine (In Vitro Technologies) and laminin (Life Technologies, Australia). The culture was maintained under standard conditions for 7 days to allow neurites to develop, before the ganglia were paraformaldehyde-fixed and stained with β-III tubulin antibody (rabbit polyclonal, 802001, BioLegend, 1:1000 in PBS with 10% FCS). The ganglia and the attached neurites were visualized by florescence-labeled β-III tubulin through FITC conjugated secondary antibody (Anti-rabbit IgG F(ab') Fragment (Alexa Fluor 488 Conjugate), 4412 S, Cell Signalling, 1:2000 in PBS with 10% FCS). The images were scanned using cell imaging system Celldiscoverer 7 (Zeiss) and analysed using ImageJ program. The fluorescence signals were converted into pixels for quantitation of the length and density of the neurites. Images were segmented using an adaptive thresholding algorithm that identifies structures that display distinct patterns in pixel intensity distinguishable from noise. A central region of interest was selected around the ganglion. Zones were constructed with a width of 300 pixels (274 μm), expanding out radially as a measure to describe positive area coverage as a function of distance from the ganglion. Measurements were normalized to area and reported as percentage area coverage.

### NMR and peptide binding analysis

Three peptides derived from human RCOR3 (residues 410–422, 428–440, and 448–460, designated RCOR3_1, RCOR3_2, and RCOR3_3 respectively), as well as one from human NCOR2 (residues 1101–1113) were purchased as N-terminally acetylated and C-terminally amidated at >80% purity from Ontores Biotechnologies. Sequences of all four peptides are shown in Fig. 6d. Peptides were prepared in MQW and their concentration was determined by A205 nm[74].

$^{15}$N HSQC spectra of 60 μm $^{15}$N-labeled NHR4 MYND were acquired in either 20 mM Mes, 100 mM NaCl, 120 μM ZnAc, 2 mM TCEP pH 6, or 20 mM Bis-Tris, 100 mM NaCl, 120 μM ZnAc, 2 mM TCEP pH 7.5. For all NMR experiments, 10% (v/v) $D_2O$ was added directly to the samples. 2,2-dimethyl-2-silapentane-5-sulfonic acid was added as a chemical shift reference to all samples to a final concentration of 500 μm. Following quantification, peptides RCOR3_1, RCOR3_2, RCOR3_3, and NCOR2 were diluted into 20 mM Mes, 100 mM NaCl, 120 μM ZnAc, 2 mM TCEP pH 6 (equivalent buffer to the protein sample) and were titrated in increments of 0.5 molar equivalents (up to 12 molar equivalents). $^{15}$N HSQC spectra were acquired after each addition. RCOR3_3 was additionally diluted into 20 mM Bis-Tris, 100 mM NaCl, 120 μM ZnAc, 2 mM TCEP pH 7.5 (equivalent buffer to the protein sample) and again titrated in increments of 0.5 molar equivalents (up to 10 molar equivalents) with $^{15}$N HSQC spectra acquired after each addition. Data were acquired at 298 K on a Bruker Avance III 600 MHz NMR spectrometer fitted with a cryogenic TCI probe head. The pulse program used for the HSQC spectra was sfhmqcf3gpph, the spectral widths were 14 ppm ($^1$H) and 32 ppm ($^{15}$N), the number of scans used was 16, and the total acquisition time was 10 min. The spectra were processed and analysed using TOPSPIN3 (version 3.6.4, Bruker, Karlsruhe, Germany).

Chemical shift perturbation (CSP) values were calculated as a weighted average of changes in $^1H$ and $^{15}N$ chemical shift between free and peptide-saturated NHR4 MYND, using Eq. 1[75], where $\Delta\delta HN$ is the chemical shift change in the proton dimension, and $\Delta\delta N$ is the chemical shift change in the nitrogen dimension. Each experiment was carried out once, and binding curves were derived by tracking the CSP values from three residues in each HSQC in response to peptide addition. The values were fitted to a simple 1:1 binding model in GraphPad Prism, and the affinities reported are the average ± SD of the three sets of measurements.

$$\Delta\delta = \sqrt{(\Delta\delta HN)^2 + (0.154 \cdot \Delta\delta N)^2} \qquad (1)$$

### Cell culture

Cell lines were mycoplasma tested and verified by short tandem repeat profiling (CellBank Australia, Westmead Australia) and were cultured under standard conditions at 37 °C, 5% $CO_2$ in Dulbecco's Modified Eagles Medium with 10% fetal bovine serum (FBS; Thermo Fisher Scientific, Australia) for BE(2)-C (ECACC Cat#95011817), SH-EP and SH-SY5Y (ECACC Cat#94030304), or Roswell Park Memorial Institute medium with 10% FBS for KELLY (ECACC Cat#92110411), DMS-273 (ECACC Cat#95062830), DMS-53 (ECACC Cat#95062823), Rh41 and Rh3 (obtained from Peter Houghton, Greehey Children's Cancer Research Institute, San Antonio, USA). Stable lines expressing doxycycline-inducible RUNX1T1 shRNA were generated using the siRNA sequence from Ambion™ (AM16708) with the pFH1UTG vector and packaged using the psPAX2 and pMD2.G plasmids (gift from Marco Herold, The Walter and Eliza Hall Institute of Medical Research, VIC Australia). SH-EP Tet-21/N were synthetically engineered to express MYCN[76]. MYCN expression is repressed upon the addition of doxycycline to barely detectable levels (gift from Manfred Schwab, German Cancer Research Center, Heidelberg, Germany). SH-EP pCDH MYCN cells were genetically engineered to stably overexpress MYCN with MYCN cloned into the lentiviral construct pCDH.

### Colony assays

For colony assays, cells were plated at 500 or 1000 cells per well in a six-well plate. For shRNA knock-down, 1 μg/mL doxycycline was added to the well and replaced every 48 h. Colonies were stained after 10–14 days using 0.5% crystal violet in 50% methanol and scanned using the Gel Doc XR+ System (Bio-Rad). Colony numbers were counted using ImageJ software. For live cell imaging, Rh41 cells were seeded in 6-well plate at a density of 100,000 cells per well and cultured overnight before transfection with 40 nM of either control siRNA (1027281) or RUNX1T1 siRNA#6 (04252038) and #8 (SI04351900) from Qiagen using Lipofectamine RNAiMAX Transfection Reagent (Thermo Fisher Scientific, Australia). At 72 h, media was changed, and cells were cultured and monitored using the IncuCyte S3 Live Cell Analysis System (Sartorius, USA). Live cell images were taken from nine spots per well every 12 h for 4 consecutive days, and the average confluence of each well was calculated.

### Western blot

Western blots were performed as previously described[77]. Primary antibodies for RUNX1T1 (rabbit polyclonal antibody, 15494-1-AP, Proteintech, 1:1000 in 4% skim milk in TBST), MYCN (mouse monoclonal B8.4.B, sc-53993, Santa Cruz, 1:1000 in 4% skim milk in TBST), GAPDH (mouse monoclonal G-9, sc-365062, Santa Cruz, 1:5000), HAND2 (Abcam, RabmAb Clone: EPR19451, 1:1000 in 4% skim milk in TBST), anti-FLAG M2 (Sigma Aldrich clone M2, affinity isolated antibody, 1:3000 in 4% skim milk in TBST), anti-HA (AbCAM ab9110, Rabbit Polyclonal HA tag antibody, 1:3000 in 3.5% skim milk in TBST), and ACTIN (rabbit polyclonal antibody, A2066, Sigma-Aldrich, 1:5000).

Western blots were imaged using the ChemiDoc™ XRS+ Imaging System and the Image Lab software (Bio-Rad).

### Protein stability

Protein synthesis of RUNX1T1 was blocked using 100 μg/mL cycloheximide (Sigma) for up to 12 h. Cells were lysed and RUNX1T1 protein levels analysed by western blotting. Densitometry from three independent experiments was used to estimate RUNX1T1 expression relative to untreated cells and is expressed as the mean ± standard error. Half-life is expressed as a percentage compared to the 0 timepoint and GAPDH was used as a loading control.

### Patient RNAseq dataset

An RNAseq dataset from the Sequencing Quality Control (SEQC) consortium consisting of 498 primary neuroblastoma samples was investigated[78]. Clinical data were available for most patients. There were 92 patients with *MYCN* amplification. This dataset can be accessed from the Gene Expression Omnibus database using accession number GSE62564.

### cDNA and PCR

For RNA, tissue samples and cells were extracted in TRIzol™ (Invitrogen) according to manufacturers' instructions. RNA was reverse transcribed using Moloney murine leukemia virus (MMLV) reverse transcriptase (Life Technologies). Gene expression was determined using SYBR green on a Bio-Rad C1000 Thermal Cycler using the primers listed in Supplementary Table 3.

Triplicate values were obtained for each sample and the amplification threshold value (Ct) defined for each assay. For determination of the expression of each gene, the comparative threshold cycle method, known as the ΔΔCt method[79], was used to quantitate target gene expression normalized relative to the control gene, to produce relative quantification.

### Luciferase assay

Murine *Runx1t1* 5′UTRs were amplified by PCR from day 0 *Th-MYCN* mice using primers listed in Supplementary Table 4 and Herculase II DNA polymerase. The PCR product was inserted into PGL4.54-LUCTK. The constructs were transiently transfected into either SH-EP pCDH (empty vector) or SH-EP pCDH MYCN cells using Lipofectamine LTX reagent (Life Technologies) following manufacturer's instructions for 48 h. Renilla-TK was used as a control, and the activity of Renilla (firefly) luciferase was measured using the Dual Luciferase Assay kit (Promega). The signal is calculated as the ratio between Firefly over Renilla (F/R). Each ratio was normalized to the empty vector control (SH-EP pCDH).

### Cloning of the MYND domain

The MYND domain of RUNX1T1[12] was cloned into a pGEX-6P vector using *BamHI* and *EcoRI* restriction sites and overexpressed as GST fusions. The YH mutant was generated using the Quikchange II mutagenesis kit (Agilent Technologies Australia), as per the manufacturer's instructions. The wildtype and mutant MYND domains were subject to Nuclear magnetic resonance (NMR) spectroscopy as previously described[80].

### Inducible RUNX1T1 shRNA constructs

For shRNA constructs, FH1UTG, a second-generation lentiviral vector derived from FUGW was used, that contains green fluorescent protein (GFP). Two shRNA constructs were created, namely FH1UTG (empty vector, EV), sh #1 (based on the Ambion™ siRNA sequence), and sh #2 (an shRNA designed upstream of the Ambion™ sequence). The forward and reverse primers (4 μL; 100 μM of each) containing compatible sticky ends (listed in Supplementary Table 5) were annealed in annealing buffer (150 mM NaCl; 1 mM EDTA; 10 mM Tris HCl, pH 8) for

40 min. Samples were phosphorylated using T4 PNK. The FH1UTG vector was digested using *BsmBI* (at 55 °C) and *XhoI* (at 37 °C) to produce ends that match the sticky ends in the primers. The vector and primers were ligated at 4 °C overnight and transformed using DH5 alpha Stbl3 *E. coli*. Constructs were transduced into BE(2)-C cells using packaging vectors PMD2G and psPAX2 (2:2:1) and sorted for the top 5% expressing GFP. All three constructs, FH1UTG, PMD2G, and psPAX2, were generous gifts from Marco Herold (The Walter and Eliza Hall Institute of Medical Research, VIC Australia). Oligos used for making these inducible shRNA constructs are listed in Supplementary Table 5.

### Computational modeling

The solution structure of the MYND domain of human RUNX1T1 complexed with the co-repressor SMRT (RCSB Protein Data Bank ID 2ODD; PMID 17560331[12]) was visualized using the PyMOL Molecular Graphics System (Schrödinger, LLC). The Tyr682His substitution was made using the PyMol Mutagenesis Wizard, and the rotamer of best fit selected.

### LC-MS/MS and Co-IP

**Co-IP.** HEK-293T cells were transiently transfected (PEI) with the indicated plasmids in equimolar ratio. Twenty-four hours after transfection, cells were scraped and collected in tubes for nuclei isolation. Samples were incubated with 500 μL of hypotonic buffer (HEPES 10 mM, NaCl 50 mM, sodium pyrophosphate 1 mM, sodium orthovanadate 1 mM, sodium fluoride 1 mM) supplied with Phenylmethanesulfonyl fluoride (Sigma) and Complete protease inhibitor (Roche) for 15 min at 4 °C and next, NP40 was added up to 0,2% w/v final concentration and incubated for 15 min. Nuclear pellets were obtained through 10 min centrifugation at 750 × $g$, and lysed in 100 μL of TNT buffer (Tris HCl pH 8.00 50 mM, NaCl 250 mM, sodium pyrophosphate 1 mM, sodium orthovanadate 1 mM, sodium fluoride 1 mM, Triton 1% w/v) supplied with phenylmethanesulfonyl fluoride (Sigma-Aldrich) and Complete protease inhibitor (Roche) for 30 min. Nuclear lysates were centrifugated at 20,000 × $g$ for 20 min to discard insoluble material, and supernatants (nuclear protein fractions) were collected and quantified with the BCA method (Thermo Fisher) following manufactural instructions. Next, 220 μg (1 μg/μL) of lysate was added to 450 μg of dried pre-equilibrated Dynabeads protein G (Thermo Fisher Scientific) for 1 h pre-clearing in rotation at 4 °C. About 20 μL of pre-cleared lysate was collected as an INPUT sample and the remaining 200 μL were incubated with 1.5 μg of Anti-FLAG antibody (#F3165 Sigma) for 1 h immunoprecipitation at 4 °C in rotation. Samples were added to 450 μg of dried pre-equilibrated Dynabeads protein G and were incubated in rotation at 4 °C for 1 h for pull-down. The beads were washed five times in 200 μL TNT buffer, eluted in 1xNuPAGE buffer (Thermo Fisher) at 70 °C for 10 min, and stored at −80 °C until immunoblot analysis.

For Co-Immunoprecipitation of MYCN and MAX, KELLY cells containing doxycycline-inducible shRNA to *Runx1t1* were cultured with and without doxycycline (1 μg/mL) for 72 h. Nuclear proteins were extracted for Co-IP using the same protocol described above. About 200 μg of nuclear protein from each treatment was incubated with 2 μg of Anti-MYCN antibody (sc-53993, Santa Cruz Biotechnology) or normal mouse IgG control (sc-2025, Santa Cruz Biotechnology) at 4 °C overnight. On the following day, 60 μL of Protein G Agarose (#11719416001, Roche) was added to each IP reaction for isolation of immunocomplexes. Purified protein was then subject to immunoblot analysis.

**LC-MS/MS.** The immunoprecipitated proteins (See Co-IP method paragraph for details) from 20 10-cm dishes of transiently (48 h) pCMV14-RUNX1T1-3xFLAG transfected BE(2)-C were precipitated with four volumes of acetone overnight at −20 °C. The protein pellet was washed with 1 mL of acetone and the sample was resuspended in 30 ul

of MS resuspension solution (6 M urea, 100 mM ammonium bicarbonate). Next, the sample was treated as follows: 3 ul of reduction solution (ammonium bicarbonate 10 mM; DTT 0.15 mg/mL) at 56 °C for 30 min, 1 ul of MS alkylation solution (Iodoacetamide 5.5 mM; ammonium bicarbonate 10 mM) at RT for 20 min in darkness, 0.6 ul of inactivation solution (DTT 50 mM; ammonium bicarbonate 10 mM). The sample was brought to 500 ul with 10 mM AmBiC to dilute the urea to a final concentration of 0.36 M and left to pass through a 3 K column. The sample was centrifuged for 10,000 × $g$ for 20 min until reaching a volume of about 250 ul and was further diluted with 250 ul of 10 mM AmBiC to finally have a final urea concentration of about 0.15 M. The sample was concentrated with a 3 K column to obtain a protein sample of about 30–50 ul. The concentrated proteins were collected in a new tube and digested with 3 ul of trypsin (Sigma-Aldrich) [0.4 ug/ul] and incubated at 37 °C O/N. 1/10 of the volume of 5% formic acid was added to the trypsinized proteins, and the proteins were lyophilized in SpeedVAC set to Dry-rate medium. Trypsinized samples were resuspended in 50 μl of water:acetonitrile:formic acid (95:3:2), sonicated, centrifuged, and 20 μl of this solution was injected into a UHPLC system (Ultimate 3000, Dionex−Thermo Fisher Scientific) coupled to a Q Exactive mass spectrometer (Thermo Fisher Scientific) equipped with a HESI-II ion source. Peptides were loaded into a C18 Hypersil Gold (100 ×2.1 mm ID, 1.9 μm ps) column (Thermo Fisher Scientific) thermostatted at 30 °C and separated using the following gradient of 0.1% formic acid in water (A) and acetonitrile (B): after 5 min, B% was raised to 3% in 2 min, then a linear gradient from 2% B to 27% B was applied in 59 min; B% was then increased to 90% in 4 min and kept constant for 3 min before the reconditioning step, for a total run time of 80 min. Ions were produced by ESI, positive polarity, 3.5 kV spray voltage, 270 °C capillary temperature, 30 auxiliary gas, 40 sheath gas, S-lens RF 45, probe was heated at 290 °C. The mass spectrometer was operated in data-dependent acquisition (DDA) mode, performing a 200 <m/z < 2000 Full MS scan at 70,000 resolution (at m/z 200) followed by HCD fragmentation at 28 normalized collision energy of the five most intense precursor ions (charge state z ≥ 2,17,500 resolution (at m/z 200)), with a dynamic exclusion of 10 s. Proteotypic peptides for RUNX1T1 were specified in the method inclusion list to increase the chances to have them selected for MS2 fragmentation. To improve the proteome coverage during the technical replicates analyses, the MS method was instructed to exclude the ions fragmented in the previous run(s). Raw data files were converted to mascot generic format (.mgf) using MSConvert (ref: http://www.proteowizard.org/tools/msconvert.html) and protein identification was performed using Mascot Server (Version 2.7.0, Matrix Science, UK) search engine against the Swissprot database (release 2018_05, 557'491 sequences) and a database of contaminants commonly found in proteomics experiments (cRAP, 116 sequences). The search parameters were set as follows: trypsin was selected as the enzyme with one missed cleavage allowed; carbamidomethylation (C) was specified as fixed modification while, oxidation (M) and deamidation (NQ) as variable modifications; peptide and MS/MS tolerances were 10 ppm (#13C = 1) and 0.05 Da, respectively. False discovery rate (FDR) evaluation was performed using a decoy concatenated search and results were filtered at 1% FDR for peptide-spectrum matches (PSMs) above homology, narrowing the search to the human proteome. Merged searches of the technical replicates were run through Mascot Daemon (Matrix Science, UK). STRING analysis, which uses a database of known and predicted protein-protein interactions (https://string-db.org/), was performed to define interaction networks involving RUNX1T1, and the parameters used to undertake the analysis are listed in Supplementary Fig. 5a.

### ChIP-seq and dual-step ChIP-seq

About 60 million KELLY cells were harvested and resuspended in PBS at a concentration of 1 million cells/mL. Thus, disuccinimidyl glutarate (DSG, Santa Cruz Biotechnologies) was added to reach 2 mM final

concentration starting from a freshly prepared 500 mM stock in DMSO. Samples were incubated at RT for 45 min, protected from light. Cells were washed three times in PBS, formaldehyde was added to a final concentration of 1% and incubated 10 min at RT on a wheel. To stop the cross-linking reaction, 0.5 mL of 2.5 M glycine was added and the samples were incubated 5 min at RT on wheel at 12 rpm. Thus, samples were centrifuged 5 min at 1500 rpm ($350 \times g$) at 4 °C and the pellet was washed three times in 10 mL of ice-cold PBS. Next, the pellet was resuspended in 1.5 mL of ice-cold cell lysis buffer for nuclei release (Pipes pH 8 5 mM, KCl 85 mM, NP40 0.5%, PMSF 1 mM, Complete 1X, MilliQ Water) and kept in ice for 10 min. Afterward, the samples were centrifuged at 1700×$g$ for 10 min at 4 °C, the supernatant was discarded, and 600 ul of RIPA Sonic-Buffer (NaCl 150 mM, NP40 1%, PMSF 1 mM, Complete 1X, SDS 0.5%, Tris HCl 50 mM, MilliQ Water) was added for a further incubation of 20 min. Nuclei were sonicated with $5 \times 10$ cycles: 30 "ON - 45" OFF; High Power. RIPA sonic buffer without SDS was added to each sample to bring SDS to a final concentration of 0.18%, and BSA-coated protein A or G sepharose beads (Thermo Fisher) were added for 30 min pre-clearing. 1/10 of the sample was kept as input. The pre-cleared lysate was incubated with 20 ug of antibody to each sample O/N. At this step, we used the following antibodies depending on the target protein: RUNX1T1 (Cat. No. C15310197 diagenode); LSD1 (C15410067 diagenode); Anti-CoREST3 / RCOR3 antibody (ab76921 Abcam). The next day, 40 ul of coated beads were added to the complexes and incubated in constant rotation for 1 h at RT. Next, complexes were washed five times with RIPA buffer; once with LiCl Wash Buffer (NaDoc 1%, NP40 1%, 500 mM LiCl, 100 mM Tris HCl, MilliQ Water.); and twice with TE buffer. Beads were resuspended in 70 μL of TE buffer plus 10 ul (10 ug/ul) of RNase A (Sigma-Aldrich), and the reaction was incubated for 45 min at 37 °C. De-crosslinking was performed by adding 20 ul of Proteinase K Buffer 5X and 6 ul of Proteinase K (19 mg/mL) by incubating samples overnight at 65 °C. The DNA was then purified by using the phenol/chloroform method and ethanol precipitation. Library for single strand NGS sequencing was prepared and analysed by IGA technology company.

Murine Runx1t1 Isoform 1 (NM_001111027.2) was amplified by PCR using primers and Herculase II DNA polymerase. The PCR product was inserted into p3xFLAG-CMV™-10 or -14 using the restriction enzymes *NotI* and *BamHI*, to preserve the open reading frame. The mutant form of *Runx1t1* (YH) was created by All-around PCR using p3xFLAG-CMV™-10 containing *Runx1t1*, and primers listed in Supplementary Table 6. The PCR product was phosphorylated by T4 PNK using the same PCR cycling as above, however the elongation step at 72 °C was increased to 5 min per cycle, and 50 ng of the product was self-ligated at 16 °C overnight. Deletion forms of CoREST3 were produced as reported for the mutant forms of *Runx1t1* using primers as follows:

Δ2_RCOR3_RV:5'-CTGGTTCAGAGTGGCAATAGGGG-3';
Δ1/2_RCOR3_FW:5'-CGCCCTGCTAATTCCATGCCA-3';
Δ1_RCOR3_RV: 5'-AGGCTGATTTAAAGTTGGCCGGG-3'

KELLY cells containing doxycycline-inducible shRNA to *Runx1t1* were cultured with and without treatment. After 72 h, $12 \times 10^6$ cells were fixed per condition to perform ChIP assays and sequencing for K4 monomethyl-histone H3 (H4K4me1), K4 dimethyl-histone H3 (H3K4me2), K4 trimethyl-histone H3 (H3K4me3), K27 acetyl-histone H3 (H3K27AC), and K27 trimethyl-histone H3 (H3K27me3) based on ChIP-seq method as previously described[81].

Briefly, KELLY cells were directly fixed in a medium containing 1% formaldehyde for 10 min at room temperature. Fixation was quenched with the addition of 125 mM (final concentration) glycine for 5 min at room temperature. Fixed cells were washed twice with ice-cold PBS and finally were harvested in cold PBS containing 1x cOmplete™, EDTA-free Protease Inhibitor Cocktail (Sigma-Aldrich, cat# 04693132001). Pellets of fixed cells were resuspended in SDS lysis buffer for 30 min on ice. The resulting lysates were sonicated to shear the DNA to fragment lengths of 200–500 bp. The complexes were immunoprecipitated with

antibodies specific for H3K4ME1 (Active motif #39297), H3K4me2 (Active motif # 39141), H3K4ME3 (Active Motif #39159), Histone H3K27AC (Active motif #39133), and H3K27M3 (Millipore #07-449).

For each immunoprecipitation, 10 μg of antibody was used per $2.5 \times 10^6$ cells. Input samples were processed in parallel. The antibody/protein complexes were collected by Protein A/G PLUS agarose beads (Santa Cruz sc-2003) and washed several times. The immune complexes were eluted with 1% SDS and 0.1 M NaHCO3 and samples were treated with proteinase K for 1 h. DNA was purified by phenol/chloroform extraction, ethanol precipitation, and resuspended in 15 μl H$_2$O.

In total, 2.5 ng of ChIP-DNA samples were used for library preparation at the Ramaciotti Centre for Genomics with ThruPLEX DNA prep KIT. The libraries were sequenced on Illumina NovaSeq 6000 S1 pair-end 100 bp reads to obtain average depth of 70 M reads per sample.

All ChIP-seq analysis was performed in triplicate and raw sequencing data for each individual replicate was aligned using Bowtie2 (v2.1.0) to the human genome reference (build hg38) with parameters -k 1 -q. Aligned reads were sorted and indexed with samtools (v1.9) and only reads with a minimum mapping quality of 30 were retained. Peak calling was performed using MACS2 (v2.1.1) on aligned reads with parameters -f BAM -q 0.05 -B –keep-dup auto. Narrow peak calling was performed on H3K4me2 and H3K4me3 and broad peak calling was performed on H3K4me1, H3K27ac, and H3K27me3. Overlapping peaks between replicates for each individual histone mark were identified using BED-Tools intersect (v2.26.0) with parameters -loj and a custom R script to merge overlapped replicate peaks. Overlapped peaks between histone marks were identified using BEDTools intersect with the parameters -wa -wb -sorted. Peaks were annotated in R (v4.1.2) with TxDb.Hsapiens.UCSC.hg38.knownGene and org.Hs.eg.db packages to identify genes within 5 kb, 50 kb, 100 kb, 200 kb, and 1 Mb from the start and end of each peak. All ChIP-seq peaks were overlapped using a custom R script where a 0 denoted the peak was absent for a given experiment and a 1 where the peak was present using the GenomicRanges package. Enhancer peaks were identified if a H3K4me3 peak was absent with a peak identified in the following histone marks: active enhancers H3K4me1+H3K27ac, poised enhancers H3K4me1 + H3K27me3 and primed enhancers H3K4me1 only. BigWig coverage tracks were generated using deeptools bamCoverage (v3.5.1) with a bin size of 10.

## Max ChIP assay

KELLY cells were treated for 72 h with either Doxycycline (1 μg/mL) or DMSO as control before ChIP assays were performed using a protocol described previously[82]. Approximately 8 μg of MAX antibody (ab199489, Abcam) or rabbit isotype control (Cat#10500 C, Invitrogen) was used in each ChIP. Primers for ChIP-qPCR are listed in Supplementary Table 7. ChIP-qPCR data were presented as fold enrichment, which is calculated by subtraction of non-MYCN target gene ABCA10 and normalization by IgG control.

## Statistical analyses

To determine if there was an association between overall survival and RUNX1T1 score for the patients in the TMA, Kaplan–Meier curves and log-rank tests were performed using IBM SPSS Statistics 24. The RUNX1T1 score was classified as either low or high at various cut points between the minimum score of 0 and the maximum score of 9. RUNX1T1 score was also correlated with *MYCN* status using GraphPad Prism 7 software (GraphPad, La Jolla, USA) and significance determined using a Mann–Whitney test. Fisher's exact tests were performed using SPSS Statistics 24.

For mouse studies, Kaplan–Meier plots of time to palpable tumor were generated, and a log-rank (Mantel–Cox) significance test was applied using GraphPad Prism 7 software. Two-tailed Fisher's exact tests (95% confidence interval) were used to determine differences between backcrossed populations (GraphPad Prism).

Other statistical methods are described in the relevant section.

## Reporting summary

Further information on research design is available in the Nature Portfolio Reporting Summary linked to this article.

## Data availability

The raw RNA-sequencing and ChIP-sequencing data generated in this study have been deposited in the GEO database under accession code GSE230265. The processed ChIP-sequencing and RNA-sequencing data are available at the GEO database under accession code GSE230265 and in our GitHub repository at https://github.com/CCI-bio/RUNX1T1. The ChIP-seq dataset of MYCN-amplified Neuroblastoma cell lines data as previously published by Durbin et al (https://doi.org/10.1038/s41588-018-0191-z) used in this study are available in the GEO database under accession code GSE94824. The mass spectrometry proteomics data are publicly available and have been deposited to the ProteomeXchange Consortium via the PRIDE partner repository with the dataset identifier PXD050375. The ChIP-sequencing and RNA-sequencing data can also be explored/analyzed directly via the R2 genomics analysis and visualization platform (https://r2.amc.nl). The remaining data were available within the Article, Supplementary Information or Source Data file. Source data are provided with this paper.

## Code availability

Computer code is available from GitHub under https://github.com/CCI-BIO/RUNX1T1. The ChIP-sequencing and RNA-sequencing data can also be explored/analyzed directly via the R2 genomics analysis and visualization platform (https://r2.amc.nl).

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

## Acknowledgements

This study was supported by grants to G.M.M., M.D.N., and M.H. from the National Health and Medical Research Council (APP1016699, APP1132608, and APP1083938), Medical Research Future Fund (APP2006645), Profield Foundation, Tour de Cure, Neuroblastoma Australia and by a grant to M.D.N., J.E.M., and A.J.G. from the Cancer Council NSW (RG 23-10). This work was also supported by grants to GP from the Italian Association for Research on Cancer (AIRC IG15182 and IG24341) and the University of Bologna, Italy (RFO 2016-2022). Children's Cancer Institute Australia is affiliated with the University of New South Wales Sydney and the Sydney Children's Hospital Network. The authors acknowledge the Sydney Analytical Core Research Facility at the University of Sydney for providing access to infrastructure, as well as the Garvan Histopathology Facility, the Animal Facility at Children's Cancer Institute, and Ramaciotti Centre for Genomics at UNSW, for providing technical assistance for this project. The imaging component of this study was carried out using instruments situated in and maintained by, the Katharina Gaus Light Microscopy Facility (KGLMF) at UNSW Sydney. We thank Janelle Collinge for the genotyping and genome sequencing conducted in mapping and identifying the Runx1t1 point mutation.

## Author contributions

Conceptualization, J.E.M., E.V., G.M., D.J.H., G.P., M.H., and M.D.N.; Methodology, J.E.M., E.V., G.M., C.M., A.J.G., C.X., B.T.K., and L.W.W.; Validation and Investigation, J.E.M., E.V., G.M., C.M., A.J.G., J.I.F., C.X., N.J., F.S., L.D.G., J.R.C.R., D.R.C., H.F., E.O.S., J.Keating, G.E., S.Allan, S.Alfred, F.K.K. A.C., H.W., A.J.R., B.T.K., M.S., P.D.R., E.D.G.F., W.G., L.W.W., J.P.M., F.M.G., J.Koster, G.P., M.H., and M.D.N.; Formal analysis, J.E.M., E.V., G.M., C.M., A.J.G., J.I.F., C.X., N.J., L.G.G., A.J.R., A.deW., B.T.K., L.W.W., J.K.K.L., J.P.M., F.M.G., J.Koster, G.P., M.H., and M.D.N.; Resources, J.E.M., E.V., G.M., C.M., A.J.G., N.J., E.D.G.F., J.P.M., J.K., G.P., M.H., and M.D.N.; Data curation, J.E.M., E.V., G.M., C.M., N.J., L.G., and J.Koster; Writing—original draft, J.E.M., E.V., G.M., G.P., M.H., and M.D.N.; Writing—review & editing, J.E.M., E.V., G.M., C.M., A.J.G., J.I.F., F.S., D.R.C., H.F., F.K.K., A.J.R., B.T.K., L.W.W., J.P.M., G.M.M., F.M.G., J.Koster, G.P., M.H., and M.D.N.; Supervision, J.E.M., E.V., G.M., C.M., D.J.H., G.P., M.H., and M.D.N.; Project administration, J.E.M., E.V., G.M., G.P., M.H., and M.D.N.; Funding acquisition, G.P., G.M.M., M.H., and M.D.N.

## Competing interests

The authors declare no competing interests.

## Additional information

[1]Children's Cancer Institute, Lowy Cancer Centre, UNSW Sydney, Kensington, NSW 2031, Australia. [2]School of Clinical Medicine, UNSW Sydney, Sydney, NSW, Australia. [3]Department of Pharmacy and Biotechnology, University of Bologna, 40126 Bologna, Italy. [4]Anatomical Pathology, NSW Health Pathology, Prince of Wales Hospital, Randwick, NSW, Australia. [5]School of Biomedical Engineering, University of Technology Sydney, Broadway, NSW, Australia. [6]Monash Biomedicine Discovery Institute, Monash University, Melbourne, VIC, Australia. [7]The Walter and Eliza Hall Institute of Medical Research, Parkville, VIC, Australia. [8]Sydney Analytical Core Research Facility, The University of Sydney, Sydney, NSW, Australia. [9]School of Life and Environmental Sciences, The University of Sydney, Sydney, NSW, Australia. [10]Kids Cancer Centre, Sydney Children's Hospital, Randwick, NSW, Australia. [11]Academic Medical Center, University of Amsterdam, Amsterdam, Netherlands. [12]UNSW Centre for Childhood Cancer Research, UNSW Sydney, Sydney, NSW, Australia. [13]Present address: Garvan Institute of Medical Research, Darlinghurst, NSW, Australia. [14]These authors contributed equally: Jayne E. Murray, Emanuele Valli, Giorgio Milazzo. [15]These authors jointly supervised this work: Giovanni Perini, Michelle Haber, Murray D. Norris. ✉e-mail: mnorris@ccia.unsw.edu.au

