## [Peer Review File · Nature Communications]

The transcriptional co-repressor Runx1t1 is essential for MYCN-driven neuroblastoma tumorigenesisREVIEWER COMMENTS

Reviewer #1 (Remarks to the Author):

In their manuscript titled “The transcriptional co-repressor Runx1t1 is essential for MYCN-driven neuroblastoma tumorigenesis,” Murray and colleagues present their results from a mutagenesis screen of the Th-MYCN mouse model of neuroblastoma. They identified the mutation and found that it is a single amino acid change in Runx1t1 and when heterozygous it suppresses MYCN mediated neuroblastoma tumorigenesis. This was validated with an independent mouse model and it was shown that loss of a single allele of Runx1t1 prevented the persistent sympathoadrenal rests that are part of normal development and go on to form tumors in the Th-MYCN model. They went on to show that MYCN drives increased translation of the Runx1t1 protein and does not affect transcription. In a TMA of patient tumors, there was correlation between MYCN amplification and Runx1t1 protein even though the mRNA levels were negatively correlated. Knockdown of Runx1t1 led to increase MYCN and reduction in its binding partner, MAX. There was less binding to MYCN/MAX target genes with Runx1t1 depletion suggesting that Runx1t1 is required for the full oncogenic activity of MYCN/MAX in neuroblastoma. In established neuroblastoma cell lines/xenografts, conditional knockdown of Runx1t1 led to reduced tumor growth and clonogenic growth. NMR of the mutant and wild type proteins suggested that the mutation led to disorder in the protein and protein biochemical studies suggested that the wild type protein binds to the CoREST complex. ChIP-seq of Runx1t1 found it enriched in intergenic regions and this correlated with location of CRC transcription factors in neuroblastoma. ChIP-seq was consistent with a model in which Runx1t1 represses enhancer activity to prevent tumor cell differentiation. There were also experiments on ESCs and ARMS.

Overall, this is an impressive mechanistic study that identifies Runx1t1 as an important gene in neuroblastoma. The concept that it prevents neuronal differentiation by keeping developmental enhancers in check is important and interesting. The only suggestions are to remove the ESC experiments and the ARMS experiments as they seem peripheral to the main focus of the study. Also, they should check the Depmap dependency screen for Runx1t1 as it should be a dependency in NB and possibly ARMS. Finally, it would be interesting to determine how retinoic acid signaling modulates neuronal differentiation in the cells with reduced Runx1t1. One might predict that it is accelerated. Also, what effect does knockdown of Runx1t1 have in mesenchymal neuroblastoma cells like GIMEN?

Reviewer #2 (Remarks to the Author):

Murray et al present a compelling study that identifies a potential downstream effector in MYCN-driven tumorigenesis in Runx1t1. Using the TH-MYCN GEMM of neuroblastoma, the authors perform a random mutagenesis study to identify mice that do not develop neuroblastoma tumors. Whole genome sequencing revealed a point mutation in RUNX1T1 at Y534 in a mouse that had developed a teratoma instead of neuroblastoma. Murray and colleagues then cross bred Runx1t1 heterozygous deleted mice with TH-MYCN to show that haploinsufficiency of Runx 1t1 had a significant impact in reducing tumor penetrance, promoting differentiation, and reducing proliferation. Knockdown of Runx1t1 in human MYCN-amplified neuroblastoma cell lines showed reduced growth in vitro and in vivo.

Biochemical studies suggest that MYCN promotes the translation of Runx1t1 and not transcription or protein stability. The authors further show that Runx1t1 participates in an epigenetic repressor complex involving COREST, HDAC and LSD1. Specifically, Runx1t1 interacts with HAND2, an important transcription factor in the neuroblastoma core regulatory circuitry. Lastly, the authors show that Runx1t1 is also important for other tumor types including MYCN-driven rhabdomyosarcoma and small cell lung carcinoma. Overall the study is well done and presents fascinating insight into the mechanism of MYCN-dependent transformation. Some clarification and additional experiments would further strengthen this study:

- 1) The initial mouse that was analyzed developed a teratoma. Why not analyze mice in which no tumors developed at all?
- 2) In several instances, the authors only look at Runx1t1 knockdown instead of R534H. Given the lack of neuroblastoma formation was observed when Runx1t1 was mutated to R534H (and not a full knockdown), several experiments should replicate the effect of the mutation to support the findings from the mutagenesis screen. These instances include:
 - a. In Fig 1D, the mice used to test Runx1t1 had a knockout of 1 allele. While R534H was tested against MYCN^{+/-} in Supp Fig 1F, it would be helpful to see how R534H affects tumorigenesis when TH-MYCN is ^{+/+} instead of ^{+/-}.
 - b. While knockdown of Runx1t1 shows a reduction in MAX protein levels, what about the Runx1t1 R534H mutant? Does that mutant also reduce MAX protein levels?
- 3) In multiple occasions, the authors make claims about histology data that appears to be the opposite conclusion when I view the data.
 - a. Figure 5E, the Kelly cells with knockdown of Runx1t1 show a morphological difference via H&E but the authors claim it does not. In particular, Supplementary Fig 4C Kelly endpoint shows cells appear larger with knockdown of Runx1t1. In other panels, the cells appear to be more spread out and the nuclei stain less prominently via H&E in Runx1t1 knockdowns. This would seem to support the findings in Fig 2C in that Runx1t1 blocks differentiation. Did a pathologist confirm that the morphology isn't different in the H&E sections?
 - b. For Supplementary Fig 4C, the authors claim there's no difference in MYCN expression at either time point. However, the endpoint looks clearly different. Loss of Runx1t1 appears to significantly downregulate MYCN protein levels which appears to contradict previous westerns (Supp Fig 3A, C). How would the authors resolve that conflict?
 - c. In Supplementary Fig 4D, again, the authors claim there's no difference in Runx1t1 or MYCN between the -dox and +dox treatments but there appears to be a significant knockdown of Runx1t1 while MYCN does appear to be reduced with the Runx1t1 knockdown. It is puzzling why the authors are claiming no differences. Isn't Runx1t1 at least expected to be reduced in +doxycycline since that has the shRNA targeting Runx1t1?
- 4) In Supplementary Fig 3A, the levels of MYCN expression appear to be increasing, even in the control shRNA conditions. Has this been repeated? If so, how do the authors explain the control shRNA increase MYCN expression? Does the second shRNA also lead to increased MYCN protein?
- 5) Figure 6B western blot should be redone to make the bands fully visible (it appears cut off in many cases). In particular, the LSD1 HA blot does not appear to show a difference in binding WT vs mutant as the authors claim. Is there additional data to suggest LSD1 is part of the complex with Runx1t1? If not, then the authors cannot make this claim. In addition, the bands appear to have some air bubbles in the bands for the CoREST3 blot which make interpretation of the results challenging.
- 6) In figure 6E, the authors claim only delta 2 showed a reduction in binding to Runx1t1. While this is definitely the strongest reduction in binding, there does appear to be reduced

binding with the delta 1 mutant as well and cannot be ignored.

7) While HAND2 is part of the neuroblastoma core regulatory circuitry, the authors don't clearly indicate how they decided to investigate the interaction between HAND2 and Runx1t1. There are other core regulatory circuitry transcription factors that could have been investigated as well. The authors should justify why they picked HAND2.

8) In Figure 7, the authors claim HAND2 recruits Runx1t1 to enhancer regions. While there is overlap in where they are binding, the authors should knockdown HAND2 to show loss of Runx1t1 at the noted sites to demonstrate HAND2 is required for Runx1t1 binding.

9) Given the main site that is mutated is a Tyrosine residue, a clear implication is that phosphorylation of that site is important for active Runx1t1. The authors can address this by immunoprecipitating cell lines with WT and R534H FLAG tagged proteins and performing a blot using the pan tyrosine phosphorylation antibody (4G10).

10) Is the mechanism of Runx1t1 mediated regulation of MAX via the PRC2 complex? The authors should show if mutant Runx1t1 alters the methylation status of the MAX promoter to solidify that link.

Reviewer #3 (Remarks to the Author):

In this manuscript, Murray, Valli, Milazzo and colleagues utilize various means of genomic, transcriptomic, and epigenomic profiling as well as functional studies in an effort to understand the role of Runx1t1 in MYCN-driven neuroblastoma tumorigenesis. They discover an exquisite dependence on Runx1t1 for tumor initiation as well as maintenance, and this appears to be related to its interaction with the LSD1-CoREST3-HDAC repressive complex via a conserved zing finger domain.

The authors present a nice trajectory starting from a screen in a mouse model and following through to functional studies in human neuroblastoma cell lines, publicly available patient data, and then extrapolate to other tumor types. There are important potential clues for the eventual application of precision medicine, but the manuscript would benefit overall from a more coordinated and thorough analysis of the data.

Major comments

1) Given that the Fig 7A-F ChIP-seq data and Fig 8A-C RNA-seq appear to have been done in the KELLY cell line, can the authors link differential regulation of enhancers in the ChIP data with RUNX1T1 KO transcriptional changes? Is there at least a subset of direct RUNX1T1 targets that can be made sense of through an integrated analysis? This might help tease apart the direct vs indirect effects of RUNX1T1-based regulation.

Related to the above, while it may or may not be a direct RUNX1T1 target, does MAX expression decrease upon RUNX1T1 knockdown?

2) The authors make a convincing argument for the MYCN-RUNX1T1 regulation happening indirectly via translation and not transcription. Still, given the documented overlap between MYCN and HAND2 binding in MYCN-amplified neuroblastomas, can the authors document whether a subset of RUNX1T1 targets are co-localizing with MYCN? They should be able to use a publicly available database such as the ChIP Atlas (<http://chip-atlas.org>) to get the MYCN binding data in at least some neuroblastoma cell lines (they do have KELLY).

3) The extrapolation of RUNX1T1 dependence to aRMS and SCLC is promising but also begs the question of overall dependence across a range of tumor types, with biomarkers potentially related to RUNX1T1 expression itself if not MYCN amplification and/or tumors containing embryonic signatures. This seems like a great opportunity to use the publicly available DepMap database to classify the dependence of RUNX1T1 across a range of tumors. A cursory exploration of the 23Q2 CRISPR data showed that one of the top hits for RUNX1T1 dependency across all cell lines was CHLA90, a neuroblastoma. Further classifications of RUNX1T1 dependency could help broaden the potential applicability of the findings in this manuscript.

4) In the western blot of RUNX1T1 in diff cell lines (e.g. MYCN-amp KELLY and BE(2)-C in Fig 5A/B as well as non-amplified SH-SY5Y in Supp Fig 4A), it appears as though different isoforms of RUNX1T1 are present (single band in KELLY, large upper band in BE(2)-C, large lower band in SH-SY5Y). Can the authors comment briefly on the relevance of the different isoforms and how studies in certain systems might be affected by this. For example, is the NHR4 domain present in all isoforms?

5) The RNA-seq data for tumor vs ganglia at 2 weeks in Fig 2D would benefit from a more thorough analysis. Even though this is a different system (Th-MYCN mice), are the genes and/or pathways altered with the heterozygous RUNX1T1 vs tumor concordant with RUNX1T1 knockdown in the human cell line data in Fig 8?

Minor comments

1) Based on the Methods section, the ChIPs for Fig 7 were all done in KELLY but should be explicitly mentioned at the beginning of the results section so the reader understands the system used.

2) Fig 7A/B seem more like supplemental data, just highlighting the genomic annotation of the three ChIP replicates. And related to basic plots for ChIP-seq data, it's a good idea to show a plot or figure that demonstrates adequate signal-to-noise ratio, perhaps a composite plot, tornado plot, or even just genome browser tracks of enriched regions.

Response to the reviewers' comments

Reviewer #1

Reviewer Comment 1.1: Suggest remove the ESC experiments and the ARMS experiments as they seem peripheral to the main focus of the study.

Response 1.1

We are not quite sure what the Reviewer is referring to by ESC experiments but assume the comments relate to the SCLC and ARMS experiments. We would prefer to leave these experiments in the manuscript (noting Reviewer #3's comment that the 'extrapolation of RUNX1T1 dependence to aRMS and SCLC is promising') as this broadens the potential role of RUNX1T1 in other cancer types and may be of particular interest to researchers working on these cancers.

Reviewer Comment 1.2: Check the Depmap dependency screen for Runx1t1 as it should be a dependency in NB and possibly ARMS.

Response 1.2

We have looked in the DepMap database for RUNX1T1 dependencies, and as shown in the graph below, hematological malignancies were the only cancer type to have a small number of cell lines showing strong dependency. These lines all have RUNX1T1 fusions which are critical in driving these cancers. After these cancers, the next highest dependency was observed in neuroblastoma, although the strength of this dependency was relatively mild, and following neuroblastoma, two out of the next three cancer types showing the highest dependencies were melanoma and SCLC, both of which, like neuroblastoma have a neural crest cell of origin. Excluding the gene fusions, there are a number of possible reasons why RUNX1T1 is not showing a stronger dependency in these cancers: i) RUNX1T1 has an important role in cell fate determination, particularly involving neuronal cell differentiation, and we found that loss of RUNX1T1 produces a differentiating effect rather than a cytotoxic effect, likely translating to weaker effects in DepMap analyses (DepMap-like assays can perform poorly when investigating differentiation therapy targets); ii) The strongest effects following RUNX1T1 downregulation that we have observed were in *Th-MYCN* mice with a fully functional immune system, and recent work (Burr *et al Cancer Cell* 36, 385–401, 2019) has shown that MYCN-amplified neuroblastomas co-opt PRC2 (a complex potently inhibited by RUNX1T1 loss) to silence MHC-I antigen processing pathway genes and enable immune evasion; iii) Dede *et al.*, (Nucl Acids Res 51, 1637–51, 2023) have provided evidence that CRISPR screens have not reached saturation in predicting gene dependencies, but rather that a typical CRISPR screen has approximately a 20% false negative rate. These false negatives occur largely within genes expressed at lower levels and RUNX1T1 is certainly in this category of expression; iv) The rhabdomyosarcoma lines have a particularly slow growth rate (RH41 60 hrs doubling time), which together with cytostatic effects may mask any effects in depmap. Taken together, there are number of possible reasons for not observing a strong dependency for RUNX1T1 in DepMap, however, it is clear from our *in vivo* mutagenesis study as well as the RUNX1T1 knockout mouse study, that this transcriptional repressor is critical for MYCN-driven neuroblastoma.

Reviewer Comment 1.3: It would be interesting to determine how retinoic acid signaling modulates neuronal differentiation in the cells with reduced Runx1t1. One might predict that it is accelerated. Also, what effect does knockdown of Runx1t1 have in mesenchymal neuroblastoma cells like GIMEN?

Response 1.3

We have now investigated the effect of RUNX1T1 knock down and retinoic acid (RA) treatment as well as their combination using BE(2)-C and GIMEN cells. As shown in the representative images below, we found that RUNX1T1 loss induced morphological differentiation (based on neurite extension) to a similar extent as to that induced by RA, however we could find no evidence for accelerated differentiation when combining both treatments. It is of interest to note that knockdown of RUNX1T1 in GIMEN cells led to growth inhibition and reduced colony formation (figure below), suggesting that this transcriptional repressor also has a role to play in maintaining the mesenchymal/undifferentiated phenotype of these cells.

Reviewer #2

Reviewer Comment 2.1: The initial mouse that was analyzed developed a teratoma. Why not analyze mice in which no tumors developed at all?

Response 2.1

Apart from the founder mouse (#1590) described in this study which developed a teratoma, the mutagenesis study yielded only one other mouse of the 1716 offspring screened that did not develop a neuroblastoma tumour. This mouse and its mutation (which is completely unrelated to RUNX1T1) are the subject of another study.

Reviewer Comment 2.2: In several instances, the authors only look at Runx1t1 knockdown instead of R534H. Given the lack of neuroblastoma formation was observed when Runx1t1 was mutated to R534H (and not a full knockdown), several experiments should replicate the effect of the mutation to support the findings from the mutagenesis screen. These instances include:

a. In Fig 1D, the mice used to test Runx1t1 had a knockout of 1 allele. While R534H was tested against MYCN \pm in Supp Fig 1F, it would be helpful to see how R534H affects tumorigenesis when TH-MYCN is \pm/\pm instead of $\pm/-$.

b. While knockdown of Runx1t1 shows a reduction in MAX protein levels, what about the Runx1t1 R534H mutant? Does that mutant also reduce MAX protein levels?

Response 2.2

a. This information is already provided in **Figure 1a** which shows the affect of the R534H mutation on tumorigenesis in Th-MYCN \pm/\pm mice. In this survival graph, the arm that is labeled 'suppressed' does in fact represent mice with the R534H mutation. The term suppressed is used here as the mutation at this point in the manuscript has not been identified.

b. This is a difficult experiment to undertake as we have no tumors or cell lines with the R534H mutation. However, we have performed transient transfections in KELLY cells using constructs containing either full length WT RUNX1T1 or the R534H mutation. We observed that overexpression of each of these constructs produced significant toxic effects with a high proportion of cells detaching from the plates. Despite this, as shown in the figure below, we did observe in the remaining cells that overexpression of the R534H mutation resulted in a reduction in MAX levels by comparison with the control cells or RUNX1T1 WT transfected cells, although this failed to achieve statistical significance ($p = 0.0796$).

Reviewer Comment 2.3: In multiple occasions, the authors make claims about histology data that appears to be the opposite conclusion when I view the data.

a. Figure 5E, the Kelly cells with knockdown of Runx1t1 show a morphological difference via H&E but the authors claim it does not. In particular, Supplementary Fig 4C Kelly endpoint shows cells appear larger with knockdown of Runx1t1. In other panels, the cells appear to be more spread out and the nuclei stain less prominently via H&E in Runx1t1 knockdowns. This would seem to support the findings in Fig 2C in that Runx1t1 blocks differentiation. Did a pathologist confirm that the morphology isn't different in the H&E sections?

Response 2.3a

All histopathologic analyses in the manuscript were performed by a formally trained and practicing pediatric pathologist (AJG) with >10 years experience.

We acknowledge that some morphologic differences are observed. Minor differences in tumor morphology (such as cell size and shape) are always present, both within the same tumor and between different tumors as well as in both control and treatment groups (in this case, doxycycline-mediated knock down). This is also the situation in both research samples and diagnostic samples in clinical practice. Variations in tissue staining (such as intensity of nuclear or cytoplasmic staining) may result from pre-analytical factors affecting tissue fixation in addition to variability in the H&E staining process itself. For this reason, we did not consider these differences to be biologically significant and we did not want to overinterpret the significance of subjective morphologic details.

It should be noted that some morphologic features are similar in both KELLY and BE(2)-C cells, in both control and knockdown tumors. For example, prominent nucleoli are evident in each of these tumor groups. It is known that prominent nucleoli in neuroblastoma cells is a morphologic correlate of *MYCN* amplification. This finding is consistent with the known *MYCN*-amplification status of both KELLY and BE(2)-C cells.

We have added further details in a new Table (**Supplementary Table 1**) summarising the morphologic assessment of these KELLY xenografts. The main text section describing these results has now been modified (page 7, line 258):

'No overall marked morphological differences were observed in KELLY tumors with RUNX1T1 knock-down and control tumors at either Day 7 or endpoint (Supplementary Table 1).'

b. For Supplementary Fig 4C, the authors claim there's no difference in MYCN expression at either time point. However, the endpoint looks clearly different. Loss of Runx1t1 appears to significantly downregulate MYCN protein levels which appears to contradict previous westerns (Supp Fig 3A, C). How would the authors resolve that conflict?

Response 2.3b

For the RUNX1T1 knock-down experiments, three cohorts of tumors were examined: KELLY Day 7, KELLY endpoint and BE(2)-C endpoint. RUNX1T1, MYCN and Ki67 protein expression was assessed, by immunohistochemistry, in two Tissue Microarrays constructed from these xenograft tumors. MYCN immunostaining was assessed in KELLY xenografts in 3 representative cores from 4 tumors at Day 7 timepoint and in 2 or 3 representative cores from 10 tumors at endpoint. Immunostaining was semi-quantitatively assessed in the form of a "TMA protein expression score" which is a number from 0 to 9, generated from the product of average intensity of staining and the % of the core with staining.

The range of protein expression scores are indicated in the below scatter plots for KELLY xenografted cells at Day 7 and endpoint. Each dot in the plot is the average of the protein expression score for an individual tumor derived from 2 or 3 individual cores taken from that tumor. The scatter plots demonstrate that there is a range of protein expression in both control and knock down tumors. Differing protein expression is also seen between cores taken from the same tumor, reflecting intratumoral variation in protein expression. No statistically significant difference in protein expression score is present for MYCN at either Day 7 ($P>0.99$) or endpoint ($P=0.58$). It is difficult for a single photo of immunohistochemical staining in a treatment group to be entirely representative of the staining observed in the tissue sections.

The absence of a significant difference in MYCN protein expression (by immunohistochemistry) in KELLY xenografts at Day 7 and endpoint, following RUNX1T1 knock-down, is not inconsistent with the observations in the Western blots in Supplementary Figure 3A and C, particularly since Supplementary Figure 3A shows the effect on MYCN following knockdown of RUNX1T1 over a short timecourse of 24hrs to 72hrs, whereas the xenograft experiments occur over a far longer period of time allowing for re-expression of RUNX1T1 and potential clonal selection. In addition, unlike the Western analysis, TMA analysis of MYCN protein expression is based on semi-quantitative assessment of immunostaining. However, we do agree with the Reviewer that visually it appears that there is a reduction of MYCN in the doxycycline-treated KELLY endpoint xenograft image by comparison with the control image (Supplementary Figure 4C; lower right panels) and so have replaced these panels with a more

representative image (based on statistical analysis) that indicates no visual difference in the levels of MYCN (**new Supplementary Figure 4C**).

Ki67 proliferative index (Ki67 PI) in KELLY has also been assessed (by visual estimation of staining in TMA cores) in tumors from both Day 7 and Endpoint. At Day 7, there is a reduction in Ki67 PI following RUNX1T1 knock-down ($47.5 \pm 4.4\%$) compared to control tumors ($80.4 \pm 4.3\%$) ($P=0.0286$). At endpoint, a small statistically significant difference in Ki67 PI is observed though it is doubtful if this is biologically significant.

These details have been added to main text section (page 7, line 254):

'Immunostaining of KELLY tumor samples excised at seven days confirmed loss of RUNX1T1 following its knockdown, as well as decreased cell proliferation as indicated by a reduced Ki67 proliferative index ($47.5 \pm 4.4\%$ in knockdown tumors compared to $80.4 \pm 4.3\%$ in control tumors; $P=0.029$) (Fig. 5G, left panels). A small reduction in Ki67 staining remained in KELLY xenografts at endpoint following RUNX1T1 knockdown ($65.0 \pm 2.1\%$) compared to control tumors (71.8 ± 1.6 ; Fig. 5G, right panels).'

The Ki67 proliferative index assessment has also been added to Methods section (page 15, line 567):

'The Ki67 proliferative index (%) was determined by visually estimating the % of viable tumour cells with Ki67 nuclear staining, to the nearest 10%, in tissue microarray cores.'

And also (page 16, line 576):

'Hyperplasia was defined as ≥ 30 neuroblasts within a ganglion, as previously described^{9,73}.'

c. In Supplementary Fig 4D, again, the authors claim there's no difference in Runx1t1 or MYCN between the -dox and +dox treatments but there appears to be a significant knockdown of Runx1t1 while MYCN does appear to be reduced with the Runx1t1 knockdown. It is puzzling why the authors are claiming no differences. Isn't Runx1t1 at least expected to be reduced in +doxycycline since that has the shRNA targeting Runx1t1?

Response 2.3c

We have now performed a more formal analysis of expression levels and on review, we agree with the Reviewer that there is a reduction in RUNX1T1 protein expression ($P=0.009$) and MYCN protein expression ($P=0.03$) in endpoint tumors following its knockdown in BE(2)-C cells. The difference in expression of MYCN, however, is minimal as shown in the plots below. RUNX1T1 and MYCN expression has been semi-quantitatively assessed from the TMAs as previously described.

We therefore have therefore updated the Results section to include this information (page 7, line 261):

'BE(2)-C cells showed reduced levels of MYCN and RUNX1T1 in doxycycline-treated samples obtained at the endpoint of the experiment compared to untreated samples. The degree of reduced expression, however, was greater for RUNX1T1 than MYCN (Supplementary Fig. 4D).'

And also in the figure legend (last sentence) to Supplementary Fig. 4:

'Reduced RUNX1T1 staining was observed in tumor cells following RUNX1T1 knock-down by comparison with the vehicle control.'

Reviewer Comment 2.4: In Supplementary Fig 3A, the levels of MYCN expression appear to be increasing, even in the control shRNA conditions. Has this been repeated? If so, how do the authors explain the control shRNA increase MYCN expression? Does the second shRNA also lead to increased MYCN protein?

Response 2.4

In Supplementary Fig 3A, the Western shows the effect that RUNX1T1 shRNA knockdown in KELLY cells has on MYCN and RUNX1T1 protein levels at three timepoints. This result has been performed in triplicate and as the graph on the right shows, despite some variability in the level of MYCN protein expression in the individual control samples, there was no significant difference in protein levels at any timepoint compared to the 0 hr timepoint. In addition, with regards the Reviewer's comment about the second RUNX1T1 shRNA (ie. shRNA#2), RNAseq analysis demonstrated a significant increase in MYCN mRNA expression in KELLY cells (1.55 fold; $p = 1.57E-14$), and as is shown in the figure below, Western analysis also confirmed increased MYCN protein levels at 48 and 72 hours after RUNX1T1 knockdown:

Reviewer Comment 2.5: Figure 6B western blot should be redone to make the bands fully visible (it appears cut off in many cases). In particular, the LSD1 HA blot does not appear to show a difference in binding WT vs mutant as the authors claim. Is there additional data to suggest LSD1 is part of the complex with Runx1t1? If not, then the authors cannot make this claim. In addition, the bands appear to have some air bubbles in the bands for the CoREST3 blot which make interpretation of the results challenging.

Response 2.5

We have now repeated all the Co-Immunoprecipitation experiments and a **new Figure 6B** has been generated with all the bands fully visible in the new immunoblot panels. On reviewing the original Figure 6B, we also agree with the reviewer that we cannot make the claim that the interaction of RUNX1T1 with LSD1 is reduced with the introduction of the point mutation and have removed this statement from the Results section (page 8, line 297). The new Figure 6B confirms there is no reduction in LSD1. Importantly, however the complete loss of interaction of mutant RUNX1T1 with CoREST3 has been confirmed with the new Co-IP data.

With regard to the reviewer's concern about whether LSD1 is part of the complex with Runx1t1, our findings are corroborated by following lines of evidence:

- 1) Our LC-MS/MS data show that LSD1 is consistently part of the complex in at least two out of three mass spec replicates (Supplementary Fig. 5A).
- 2) The interaction of LSD1 with RUNX1T1 has also been specifically observed in Co-IP assays (Fig. 6B).
- 3) We observed consistent genome wide co-localization of RUNX1T1 with LSD1 (see Figure 7D).

Given the fact that CoREST proteins are consistently associated with LSD1 protein together with HDACs (Fig. 6B), we believe that this evidence is sufficiently strong to support the conclusion that RUNX1T1 is part of a repressive complex together with LSD1.

Reviewer Comment 2.6: In figure 6E, the authors claim only delta 2 showed a reduction in binding to Runx1t1. While this is definitely the strongest reduction in binding, there does appear to be reduced binding with the delta 1 mutant as well and cannot be ignored.

Response 2.6

We agree with the reviewer that there is reduced binding with the delta 1 mutant and have therefore added the following text to the Results section (page 9, line 317):

Based on these data, two CoREST3 mutant proteins, one lacking an amino acid sequence of the last PPPLI motif ($\Delta 1$) and another lacking all three "PPPLI motifs" identified ($\Delta 2$), were generated (Fig. 6E) and Co-IP performed with RUNX1T1. Although a decrease in binding to RUNX1T1 was observed with the CoREST3_ $\Delta 1$ -mutant, binding of the CoREST3_ $\Delta 2$ -mutant protein to RUNX1T1 (Fig. 6E) was completely lost, suggesting that this region of CoREST3 is critical for the binding to RUNX1T1.'

Reviewer Comment 2.7: While HAND2 is part of the neuroblastoma core regulatory circuitry, the authors don't clearly indicate how they decided to investigate the interaction between HAND2 and Runx1t1. There are other core regulatory circuitry transcription factors that could have been investigated as well. The authors should justify why they picked HAND2.

Response 2.7

HAND2 was chosen based on the evidence from the mass spectrometry data, and subsequently confirmed using co-immunoprecipitation, showing this transcription factor formed part of a chromatin remodeling complex containing RUNX1T1, LSD1/KDM1A, RCOR3/CoREST3 and HDAC1-3 (Supplementary Fig. 5A). We then performed a *de novo* motif search using HOMER on RUNX1T1 ChIPseq peaks and the strongest alignment was with the motif for HAND2 (Fig. 7C; $p=1e-98$) and following this we also found that 36% of all RUNX1T1 ChIPseq overlapped with HAND2 peaks. However, in light of the reviewer's comments regarding 'other core regulatory circuitry transcription factors' (CRCs), we investigated the ChIPseq peaks from these transcription factors (ISL1, PHOX2B, GATA3, TBX2) making up this core circuitry (Durbin *et al*; *Nat Genet* 50, 1240-46, 2018). As shown in the Table below, there was very little overlap with any these transcription factor ChIPseq peaks with RUNX1T1 peaks (413/15612 or 2.6%) in KELLY cells, ranging from as low as 0.7% for ISL1 and up to 2.4% for PHOX2B. This result is in stark contrast to the 36% of RUNX1T1 peaks that overlap with HAND2 and further confirms the importance of HAND2 in recruiting a RUNX1T1 repressor complex to help maintain an undifferentiated phenotype.

CRC Gene	RUNX1T1 ChIPseq Peak overlap	
	No. Peaks	% of Total
ISL1	102	0.7%
PHOX2B	339	2.4%
GATA3	206	1.5%
TBX2	197	1.4%
MYCN	360	2.6%
HAND2	5073	36%

Reviewer Comment 2.8: In Figure 7, the authors claim HAND2 recruits Runx1t1 to enhancer regions. While there is overlap in where they are binding, the authors should knockdown HAND2 to show loss of Runx1t1 at the noted sites to demonstrate HAND2 is required for Runx1t1 binding.

Response 2.8

We have now performed the HAND2 knockdown experiment as requested by the reviewer, and the results show that when HAND2 is conditionally knocked down, the binding of RUNX1T1 to chromatin is significantly decreased, suggesting that HAND2 is critical for RUNX1T1 binding to chromatin. These data are summarized in three new panels shown below. Panels **A** and **B** show highly significant knockdown of HAND2 after 72hrs of doxycycline (1 μ g/mL) treatment in KELLY cells, but no significant change in the level of RUNX1T1 protein. However, following ChIP-qPCR on four randomly chosen ChIP-seq peaks that demonstrated positive binding for both HAND2 and RUNX1T1, loss of HAND2 led to a significant decrease of RUNX1T1 binding.

These results have now been added as new panels to Supplementary Figure 5, with the following description in the Results section (page 9, line 334):

'Additionally, we performed ChIP-qPCR analysis for RUNX1T1 following shRNA-mediated inducible knockdown of HAND2. Loss of HAND2 had no significant effect on RUNX1T1 expression (Supplementary Fig. 5E,F). However, ChIP-qPCR performed on four randomly chosen ChIP-seq peaks previously found to be positive for both HAND2 and RUNX1T1, demonstrated a significant decrease in RUNX1T1 binding following loss of HAND2 (Supplementary Fig. 5G).'

And also in the figure legend to Supplementary Fig. 5:

*'(E) HAND2 Knock-Down with shHAND2 after 72 hrs of doxycycline (1µg/mL) treatment in KELLY cells. (F) normalized quantification from three independent experiments demonstrated significantly decreased HAND2 levels and no significant change in RUNX1T1 following shRNA-mediated HAND2 knockdown. (G) RUNX1T1 and HAND2 ChIP-qPCR assays in KELLY cells with HAND2 Tet-inducible KD) cells following doxycycline treatment (1 µg/ml for 48 hrs). Input (white bars), HAND2 IP (light gray bars) and RUNX1T1 IP (dark grey bars) samples were analyzed by q-PCR using specific primers for Peak 1, 2, 3, 4 and ABCA10 TSS as negative control (Supplementary Table 7); dotted bars refers to HAND2 KD condition (+ doxycycline). Data are mean ± SD (n = 3 biological replicates; two-tailed unpaired t-test). * p < 0.05; ** p < 0.01; *** p < 0.001.'*

Reviewer Comment 2.9: Given the main site that is mutated is a Tyrosine residue, a clear implication is that phosphorylation of that site is important for active Runx1t1. The authors can address this by immunoprecipitating cell lines with WT and R534H FLAG tagged proteins and performing a blot using the pan tyrosine phosphorylation antibody (4G10).

Response 2.9

As suggested by the reviewer we have performed this experiment investigating tyrosine phosphorylation in RUNX1T1. HEK-293t cells were transiently transfected (24 hrs) with RUNX1T1 constructs expressing either FLAG-tagged wild type or R534H mutant protein and resulting samples were immunoblotted and probed with anti-FLAG antibody (panel A below, 20% of IP), or anti-phosphoTyr (panel B, 80% of IP), and then probed with an anti-light chain specific anti-Mouse secondary antibody. Input sample represents 5% of the non-immunoprecipitated sample. We could find no evidence of phosphorylated tyrosine residues present in either the wild type or mutant RUNX1T1 proteins (panel B showing low and high exposures). Thus, we could not find any evidence to suggest that the mutated tyrosine residue in RUNX1T1 is phosphorylated.

Reviewer Comment 2.10: Is the mechanism of Runx1t1 mediated regulation of MAX via the PRC2 complex? The authors should show if mutant Runx1t1 alters the methylation status of the MAX promoter to solidify that link.

Response 2.10

Polycomb repressive complex 2 or PRC2 is a chromatin-modifying enzyme complex whose role as its name implies is to maintain transcriptional repression of target genes. A number of reports have shown that MYCN physically interacts with PRC2 (e.g. Corvetta *et al.*, J Biol Chem, 288: 8332-8341, 2013) and this complex is required for the oncogenic potential of MYCN. Thus, RUNX1T1 cannot be mediating the regulation of MAX via PRC2, since inhibition of RUNX1T1 reverses the effects of PRC2 and would lead to increased expression of MAX rather than the decreased expression that we have observed. Similarly, with regards MAX promoter methylation, our ChIPseq data shows that RUNX1T1 binds almost exclusively to intergenic regions and there is no evidence of binding to the MAX promoter.

Reviewer #3

Reviewer Comment 3.1: Given that the Fig 7A-F ChIP-seq data and Fig 8A-C RNA-seq appear to have been done in the KELLY cell line, can the authors link differential regulation of enhancers in the ChIP data with RUNX1T1 KO transcriptional changes? Is there at least a subset of direct RUNX1T1 targets that can be made sense of through an integrated analysis? This might help tease apart the direct vs indirect effects of RUNX1T1-based regulation.

Related to the above, while it may or may not be a direct RUNX1T1 target, does MAX expression decrease upon RUNX1T1 knockdown?

Response 3.1

Integrated analysis: In order to compare the concordance of enrichment between the ChIPseq data with the RNAseq data from the KELLY cells, we firstly extracted the genes associated with the GREAT analysis performed on the 24% of common peaks co-localizing with RUNX1T1, HAND2, LSD1 and CoREST3, and then extracted the differential expression data from the three RNAseq experiments (Kelly cell lines and PDX engrafted mice) comparing RUNX1T1 knock-out against control (Figure 8A). Enrichment analysis was then performed on the dataset containing genes common to both extractions following ranking using false discovery rate (FDR) and fold-change values. As detailed in the figure below, gene ontology analysis showed that the top upregulated biological processes included sympathetic nervous system development and neuron fate specification. Such neuronal development processes, including sympathetic nervous system development, were also among the top processes observed following GREAT analysis done on the ChIPseq RUNX1T1 peaks co-localizing with HAND2, LSD1 and CoREST3 (Figure 7E), suggesting a high degree of concordance between these two datasets. However, although the biological processes in the figure below were statistically significant ($p < 0.05$), after benjamini multiple test correction they failed to reach significance. For this reason we are not proposing to include this figure in the manuscript.

MAX expression: With regards the Reviewer's question of whether MAX expression decreases upon RUNX1T1 knockdown, we have observed highly significant downregulation (1.6×10^{-14}) of MAX mRNA in

the RNAseq data following shRNA-mediated knockdown of RUNX1T1 in KELLY cells, and as shown in Supplementary Figure 3B, a corresponding >50% reduction in the level of MAX protein.

Reviewer Comment 3.2: The authors make a convincing argument for the MYCN-RUNX1T1 regulation happening indirectly via translation and not transcription. Still, given the documented overlap between MYCN and HAND2 binding in MYCN-amplified neuroblastomas, can the authors document whether a subset of RUNX1T1 targets are co-localizing with MYCN? They should be able to use a publicly available database such as the CHIP Atlas (<http://chip-atlas.org>) to get the MYCN binding data in at least some neuroblastoma cell lines (they do have KELLY).

Response 3.2

As requested by the Reviewer, we have now determined the overlap between RUNX1T1 ChIPseq peaks and MYCN ChIPseq peaks previously reported in KELLY cells by Durbin *et al* (*Nat Genet* 50, 1240-46, 2018). This analysis is presented below and shows that only 360 RUNX1T1 peaks or 2.6% (360 from a total of 14019 RUNX1T1 peaks) overlapped with MYCN peaks, while only 219 RUNX1T1 peaks (1.6%) overlapped with MYCN, HAND2, LSD1 and CoREST3. This represents a small number of RUNX1T1 peaks by comparison with the 5073 peaks (36%) that overlapped with HAND2. We have also shown that there is very little overlap of RUNX1T1 peaks with the other members of the Core Regulatory Circuitry described by Durbin *et al* (see Response 2.7), thus further highlighting the role of HAND2 in maintaining an undifferentiated phenotype in MYCN-driven neuroblastoma cells.

This information has now been included in a new Figure 7D with the following description added to main text section (page 9, line 342):

'These findings are in marked contrast with those for the MYCN ChIP-seq dataset from KELLY cells²⁶, where 2.6% of RUNX1T1 ChIP-seq peaks overlapped with MYCN peaks, while only 1.6% of RUNX1T1 peaks co-localized with MYCN, HAND2, LSD1 and CoREST3 (Fig. 7D). We next used Genomic Regions Enrichment of Annotations Tool (GREAT) analysis to explore gene ontology processes associated with the 24% of common peaks co-localizing with RUNX1T1, HAND2, LSD1 and CoREST3.'

Reviewer Comment 3.3: The extrapolation of RUNX1T1 dependence to aRMS and SCLC is promising but also begs the question of overall dependence across a range of tumor types, with biomarkers potentially related to RUNX1T1 expression itself if not MYCN amplification and/or tumors containing embryonic signatures. This seems like a great opportunity to use the publicly available

DepMap database to classify the dependence of RUNX1T1 across a range of tumors. A cursory exploration of the 23Q2 CRISPR data showed that one of the top hits for RUNX1T1 dependency across all cell lines was CHLA90, a neuroblastoma. Further classifications of RUNX1T1 dependency could help broaden the potential applicability of the findings in this manuscript.

Response 3.3

Please see response to Reviewer comment 1.2.

Reviewer Comment 3.4: In the western blot of RUNX1T1 in diff cell lines (e.g. MYCN-amp KELLY and BE(2)-C in Fig 5A/B as well as non-amplified SH-SY5Y in Supp Fig 4A), it appears as though different isoforms of RUNX1T1 are present (single band in KELLY, large upper band in BE(2)-C, large lower band in SH-SY5Y). Can the authors comment briefly on the relevance of the different isoforms and how studies in certain systems might be affected by this. For example, is the NHR4 domain present in all isoforms?

Response 3.4

We do occasionally see additional bands in our RUNX1T1 Western blots, which we believe are non-specific in nature. These additional bands are most likely due to the use of the polyclonal RUNX1T1 antibody employed in these experiments, since we have recently obtained a RUNX1T1 monoclonal antibody, and as shown in the Western blots below, a single band or isoform is routinely observed in the three cell lines mentioned i.e. KELLY, BE(2)-C and Sh-SY5Y cells. Additionally, in the literature two RUNX1T1 isoforms have been reported that encode either a full length protein or a shorter isoform, lacking exon 6 (*Cell Research* 24:1403 -19, 2014). During adipogenesis, the two isoforms have competing functions with the long isoform inhibiting adipocyte differentiation and the short isoform promoting differentiation. Our analysis of RNAseq data from a large number of primary neuroblastoma tumors found no evidence of the RUNX1T1 short isoform, which is in keeping with our finding that full length RUNX1T1 isoform is preventing neuroblastoma differentiation.

Reviewer Comment 3.5: The RNA-seq data for tumor vs ganglia at 2 weeks in Fig 2D would benefit from a more thorough analysis. Even though this is a different system (Th-MYCN mice), are the genes and/or pathways altered with the heterozygous RUNX1T1 vs tumor concordant with RUNX1T1 knockdown in the human cell line data in Fig 8?

Response 3.5

As requested by the Reviewer, we have now performed Gene Set Enrichment Analysis of Hallmarks on the ganglia obtained at two weeks from Runx1t1 heterozygous mice (Th-MYCN^{+/+}; Runx1t1^{+/-}) versus tumor material from homozygous mice (Th-MYCN^{+/+}; Runx1t1^{+/+}). The results are highly concordant with the human GSEA data in Figure 8A, showing the same top upregulated and downregulated Hallmark gene sets, despite this being a comparison between human and murine cells.

This information has now been included in a new Figure 8A with the following description added to main text section (page 10, line 381):

‘Furthermore, when we included in this analysis RNAseq data of ganglia (Th-MYCN^{+/+}; Runx1t1^{+/-}) and tumor (Th-MYCN^{+/+}; Runx1t1^{+/+}) from our neuroblastoma-prone mice, GSEA showed highly concordant results with the human data, with enrichment of the same top upregulated and downregulated Hallmark gene sets in mouse ganglia carrying only one function Runx1t1 allele (Fig. 8A).’

The following information has also been included in the legend to Figure 8:

‘The column labelled ganglia represents the same analysis performed on ganglia (Th-MYCN^{+/+}; Runx1t1^{+/-}) at two weeks versus tumor obtained from Th-MYCN^{+/+}; Runx1t1^{+/+} mice.’

Reviewer minor Comment 3.1: Based on the Methods section, the ChIPs for Fig 7 were all done in KELLY but should be explicitly mentioned at the beginning of the results section so the reader understands the system used.

Response 3.1

The cell line used has now been added to the Results (page 9, line 326)::

‘...we performed chromatin immunoprecipitation and sequencing (ChIP-seq) on KELLY cells.’

Reviewer minor Comment 3.2: Fig 7A/B seem more like supplemental data, just highlighting the genomic annotation of the three ChIP replicates. And related to basic plots for ChIP-seq data, it's a good idea to show a plot or figure that demonstrates adequate signal-to-noise ratio, perhaps a composite plot, tornado plot, or even just genome browser tracks of enriched regions.

Response 3.2

Figure 7A and 7B were selected as main panel figures to emphasise the fact that RUNX1T1 binds to intergenic regions. We regard this as important information as it emphasizes the fact that RUNX1T1 doesn't bind to transcription start sites/promoters of genes which directly contrasts with transcription factors such as MYCN. Although ChIP-seq experiments typically show super-enhancers or the effect of transcription factor binding within selected regions of the genome, this is not the case here. We would therefore prefer to leave these panels where they are as we want to highlight the distribution of the binding locations across the genome. However, the heatmaps of the 3 RUNX1T1 ChIP-seq replicates are shown below to show the quality and reproducibility of the data generated.

REVIEWERS' COMMENTS

Reviewer #2 (Remarks to the Author):

The authors have sufficiently addressed my concerns

Reviewer #3 (Remarks to the Author):

The authors have sufficiently answered my questions and satisfied all concerns, and I thank them for the thoughtful and detailed responses. I have no further concerns and believe the manuscript will be of interest to the field.